# Provable Scaling Laws for the Test-Time Compute of Large Language Models

**Yanxi Chen**[*]
Alibaba Group
chenyanxi.cyx@alibaba-inc.com

**Xuchen Pan**[*]
Alibaba Group
panxuchen.pxc@alibaba-inc.com

**Yaliang Li**
Alibaba Group
yaliang.li@alibaba-inc.com

**Bolin Ding**
Alibaba Group
bolin.ding@alibaba-inc.com

**Jingren Zhou**
Alibaba Group
jingren.zhou@alibaba-inc.com

## Abstract

We propose two simple, principled and practical algorithms that enjoy provable scaling laws for the test-time compute of large language models (LLMs). The first one is a two-stage knockout-style algorithm: given an input problem, it first generates multiple candidate solutions, and then aggregate them via a knockout tournament for the final output. Assuming that the LLM can generate a correct solution with non-zero probability and do better than a random guess in comparing a pair of correct and incorrect solutions, we prove theoretically that the failure probability of this algorithm decays to zero exponentially or by a power law (depending on the specific way of scaling) as its test-time compute grows. The second one is a two-stage league-style algorithm, where each candidate is evaluated by its average win rate against multiple opponents, rather than eliminated upon loss to a single opponent. Under analogous but more robust assumptions, we prove that its failure probability also decays to zero exponentially with more test-time compute. Both algorithms require a black-box LLM and nothing else (e.g., no verifier or reward model) for a minimalistic implementation, which makes them appealing for practical applications and easy to adapt for different tasks. Through extensive experiments with diverse models and datasets, we validate the proposed theories and demonstrate the outstanding scaling properties of both algorithms.

## 1 Introduction

Despite the astonishing advancements of large language models (LLMs) in the past few years, they still face challenges with reliability and stability. This hinders their applications in high-stakes scenarios where a problem need to be solved with success probability $99.9\%$ rather than $90\%$. Similarly, in an LLM-based agentic workflow that involves solving many sub-problems, each of them need to be solved with near-perfect success rate, since a single error in the process can lead to an incorrect final output. In these and many other similar scenarios, one is willing to boost the success probability by spending more test-time compute on LLM inference. One category of methods for this purpose include iterative approaches like generating a sequential chain of thoughts

---

[*]Equal contributions.

39th Conference on Neural Information Processing Systems (NeurIPS 2025).

[43, 18, 29, 53, 7, 37] or self-refinement [6, 25, 8, 50, 51]. Another category, which is the focus of this work, is about repeatedly sampling multiple solutions and then aggregating them for the final output; examples include best-of-N sampling [5, 2, 36, 55, 34], majority voting [41, 3, 21], among others [13, 15, 22, 47, 54]. These two categories are complementary and often used together for the best performance [35, 44, 53, 12, 28].

**Goal of this work.** We aim to augment the toolkit of LLM inference scaling with practical algorithms and foundational insights. Throughout this work, we consider a generic problem formulation where an LLM-based algorithm is given an input problem and asked to output a solution. For conceptual simplicity, we evaluate any solution with a binary metric indicating whether it is correct or incorrect. We desire algorithms that enjoy provable inference scaling laws in the following sense:

**Definition 1.1.** We say that an LLM-based algorithm enjoys a *provable inference scaling law* for a specific input problem, if its success probability (with respect to the inherent randomness of the algorithm) in returning a correct solution to the problem can be boosted arbitrarily close to $100\%$ as its test-time compute grows, provided that certain clearly identified assumptions about the problem and the LLM(s) being used are satisfied.

**Limitations of existing methods.** Strong baseline methods widely adopted in practice may fail (in theory and practice) to achieve this goal, even if a single LLM call already solves the input problem correctly with high or moderate success probability. For example, best-of-N (BoN) sampling with an imperfect verifier might suffer from performance decay as the number of sampled solutions grows because, as explained in Section 5.1 of [5], "the benefits of search are eventually outweighed by the risk of finding adversarial solutions that fool the verifier". Indeed, prior works highlighted that developing test-time scaling methods that do not rely on perfect verifiers remains an important direction for further research [2, 36]. Majority voting, another strong baseline, might fail for different reasons: even if the LLM has moderate probability, say 45%, of generating a correct final answer, the success probability of majority voting will actually converge to *zero* as the number of samples grow if there exists an incorrect final answer that a single LLM call generates with probability 46% [3].

**Main contributions.** In pursuit of provable inference scaling laws, we propose a *two-stage knockout-style algorithm* that first generates multiple candidate solutions, and then select one via a knockout tournament where pairwise comparisons among the candidates are conducted. We prove theoretically that its failure probability *decays to zero exponentially* (Theorem 2.3) or *by a power law* (Theorem 2.4) with respect to the total number of LLM calls, depending on the specific way of scaling. These guarantees rely on two assumptions: (1) the LLM can generate a correct solution with *non-zero* probability, and (2) the LLM can *do better than a random guess* in choosing the right winner between any pair of correct and incorrect solutions.

We further propose a *two-stage league-style algorithm* that also enjoys a provable scaling law. Unlike the knockout-style algorithm that eliminates a candidate upon loss to a single opponent, the league-style algorithm evaluates each candidate by its average win rate against multiple opponents. We prove that its failure probability also *decays to zero exponentially* (Theorem 3.3) as its test-time compute scales up, under the technical assumption that there exist correct solutions whose average win rates against a distribution of opponents are higher than that of any incorrect solution.

Both proposed algorithms require a black-box LLM and nothing else (e.g., no external verifier or reward model) for a minimalistic implementation, which makes them appealing for practical applications and easy to adapt for different scenarios. Our practical implementations are efficient and scalable, with support for parallel and distributed computation. While the technical assumptions in our theories might seem strong from a practical perspective, our empirical results confirm that the proposed algorithms — developed based on the theoretical insights — indeed perform well and demonstrate outstanding scaling properties across diverse LLMs (`Llama3.1`, `Qwen2.5`, `GPT-4o`, `QwQ-32B`) and datasets (GPQA, MMLU-Pro, MATH-500).

## 2    A two-stage knockout-style algorithm

This section studies the following two-stage knockout-style algorithm for solving an input problem:

1. *Generation.* We first generate $N$ candidate solutions, which can run in parallel. In situations where the final answer contains only a few tokens (e.g., for multiple-choice problems or math calculation), we require that each solution contains a thinking process, which can be elicited by chain-of-thought (CoT) prompting [43, 18] for example; such information can be useful for enhancing pairwise comparisons in the next stage.

2. *Aggregation.* We aggregate the candidate solutions via a knockout tournament. At each round, the candidates are grouped into pairs randomly, and each pair of candidates are compared for $K$ times. The winner of each pair is the one that is favored for more than $K/2$ times; ties are broken randomly. Only the winners will move on to the next round. The final-round winner at the end of this tournament will be the final output of the algorithm.

Figure 1 visualizes this process. For a minimalistic implementation, both stages can be executed with a single black-box LLM (or an ensemble of multiple LLMs). We next introduce some formal notations, followed by our analysis for both success probability and computational efficiency [16, 4] of the proposed algorithm, to be presented in Sections 2.1 and 2.2 respectively.

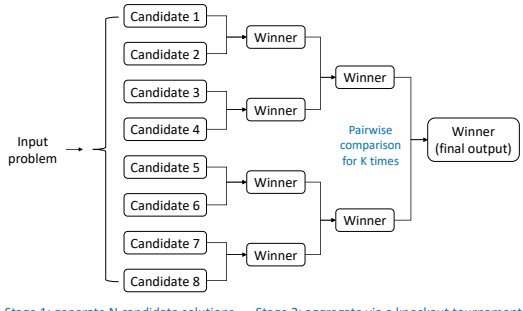

Figure 1: A visualization of the proposed two-stage knockout-style algorithm, with $N = 8$ in this example.

**Formal notations.** Let $\mathcal{M}_{\mathrm{gen}}$ and $\mathcal{M}_{\mathrm{comp}}$ denote the probability distribution of the output of one LLM call for generating a solution and for comparing a pair of solutions respectively. Given an input problem $x$, the proposed algorithm first samples $N$ independent candidate solutions $y_1, \ldots, y_N \sim \mathcal{M}_{\mathrm{gen}}(x)$ during the generation stage. Then, for each pair of candidates $(y, y')$ encountered in the knockout stage, the algorithm samples $K$ independent comparison results $r_1, \ldots, r_K \sim \mathcal{M}_{\mathrm{comp}}(x, y, y')$, and identifies the candidate that is favored by the majority of $\{r_i\}_{i \in [K]}$ as the winner. Sampling from $\mathcal{M}_{\mathrm{gen}}$ and $\mathcal{M}_{\mathrm{comp}}$ throughout the algorithm is the sole source of randomness in the following analysis of success probability. The randomness within $\mathcal{M}_{\mathrm{gen}}$ and $\mathcal{M}_{\mathrm{comp}}$ can originate from LLM decoding with a non-zero temperature, the randomized choice of prompting method or LLM backend for each LLM call, among others.

## 2.1 Analysis of success probability

Our theoretical guarantees for the proposed algorithm rely on the following assumption about the input problem under consideration and the LLM(s) being used.

**Assumption 2.1.** For the input problem $x$, there exists $p_{\mathrm{gen}} > 0$ such that

$$\mathbb{P}_{y \sim \mathcal{M}_{\mathrm{gen}}(x)}(y \text{ is a correct solution}) \geq p_{\mathrm{gen}} > 0.$$

In addition, there exists $p_{\mathrm{comp}} > 0.5$ such that, for an arbitrary pair of candidate solutions $(y, y')$ where one of them is correct and the other is incorrect, it holds that

$$\mathbb{P}_{r \sim \mathcal{M}_{\mathrm{comp}}(x, y, y')}(r \text{ identifies the right winner}) \geq p_{\mathrm{comp}} > 0.5.$$

In other words, we assume that the LLM can generate a correct solution with non-zero probability, and do better than a random guess in comparing a pair of correct and incorrect solutions. Here, $p_{\mathrm{gen}}$ and $p_{\mathrm{comp}}$ are defined for a specific input problem, not for a distribution of problems or a benchmark.

*Remark* 2.2. While this assumption seems minimal and natural at first glance, its requirement that $p_{\mathrm{comp}} > 0.5$ holds for *any pair* of correct and incorrect solutions renders it somewhat restricted and non-robust. This will motivate our development for an alternative algorithm and its provable scaling law under different technical assumptions, to be elaborated in Section 3.

**Scaling up both $N$ and $K$.** As $N$ (the number of initial candidate solutions) and $K$ (the number of times that each pair of solutions involved in the knockout stage are compared) grow, it becomes

more likely that (1) there exist initial candidate solutions that are correct ones, and (2) they tend to be selected as the winners in pairwise comparisons against incorrect solutions, which together lead to the correctness of the final output of the algorithm. This is formalized in the following theorem.

**Theorem 2.3.** *If Assumption 2.1 holds for the input problem, then the probability that the proposed knockout-style algorithm returns an incorrect final output decays to zero exponentially with respect to the hyperparameters $N$ and $K$:*

$$\mathbb{P}(\textit{failure}) \leq (1 - p_{gen})^N + \lceil \log_2 N \rceil e^{-2K(p_{comp}-0.5)^2}.$$

A proof can be found in Appendix B.1. Another way to interpret this theorem is as follows: for a targeted success probability $1 - \delta$ (which can be arbitrarily close to 1 as the failure probability $\delta > 0$ approaches zero), it suffices to have

$$N \geq \frac{1}{p_{\text{gen}}} \log\left(\frac{2}{\delta}\right) \quad \text{and} \quad K \geq \frac{1}{2(p_{\text{comp}} - 0.5)^2} \log\left(\frac{2\lceil \log_2 N \rceil}{\delta}\right). \tag{1}$$

In other words, $N$ and $K$ have logarithmic dependence on $1/\delta$, and linear dependence on $1/p_{\text{gen}}$ and $1/(p_{\text{comp}} - 0.5)^2$ respectively.

**Scaling up $N$ while $K$ is fixed.** Although Theorem 2.3 guarantees an arbitrarily small failure probability, it requires $K$ to be sufficiently large, depending on the value of $p_{\text{comp}}$ that might be unknown a priori in practice. To resolve this, we provide an alternative theorem suggesting that scaling up $N$ alone is sufficient, even when $K$ is a fixed constant and thus there is still a good chance that the wrong winner is identified when comparing a specific pair of candidates.

To streamline the statement of our theorem, we introduce the notations $\mathcal{M}_{\text{comp},K}$ and $p_{\text{comp},K}$, which generalize $\mathcal{M}_{\text{comp}}$ and $p_{\text{comp}}$ that appear in Assumption 2.1. Let $\mathcal{M}_{\text{comp},K}$ denote the probability distribution of the comparison result obtained with $K$ independent LLM calls followed by majority voting (with ties broken randomly). Then we have

$$\mathbb{P}_{r \sim \mathcal{M}_{\text{comp},K}(x,y,y')}(r \text{ identifies the right winner}) \geq p_{\text{comp},K} > 0.5,$$

where $p_{\text{comp},K} = \sum_{\ell=\lceil K/2 \rceil}^{K} \binom{K}{\ell} p_{\text{comp}}^\ell (1 - p_{\text{comp}})^{K-\ell}$ if $K$ is odd, and $p_{\text{comp},K} = \sum_{\ell=K/2+1}^{K} \binom{K}{\ell} p_{\text{comp}}^\ell (1 - p_{\text{comp}})^{K-\ell} + \frac{1}{2}\binom{K}{K/2} p_{\text{comp}}^{K/2} (1 - p_{\text{comp}})^{K/2}$ if $K$ is even.

**Theorem 2.4.** *Suppose that Assumption 2.1 holds and $N$ is a power of 2. Let $p_i$ be the probability that a candidate solution at the $i$-th level of the knockout tournament is correct, where $i = 0, 1, \ldots, \log_2 N$.[2] Then $p_0 = p_{gen}$ and*

$$p_{i+1} \geq p_i + (2p_{comp,K} - 1)(p_i - p_i^2) \quad \textit{for} \quad i = 0, 1, \ldots, \log_2 N - 1.$$

*Consequently, the success probability of the overall algorithm, namely $p_{\log_2 N}$, converges to 1 as $N$ grows; for any $0 < \delta < 0.5$, one has $p_{\log_2 N} \geq 1 - \delta$ as long as*

$$\log_2 N \geq \frac{\log\left(\max\{\frac{1}{2p_{gen}}, 1\}\right)}{\log\left(1 + (p_{comp,K} - 0.5)\right)} + \frac{\log\left(\frac{1}{2\delta}\right)}{-\log\left(1 - (p_{comp,K} - 0.5)\right)}.$$

A proof can be found in Appendix B.2. The linear relationship between $\log_2 N$ and $\log(1/\delta)$ reveals a *power-law* relationship between the failure probability $\delta$ and the number of candidates $N$.

## 2.2 Analysis of computational efficiency

The minimalistic implementation of the proposed knockout-style algorithm starts by generating $N$ candidate solutions with $N$ LLM calls that can run in parallel. Since the number of candidates is reduced by half at each round of the knockout tournament, there is at most $\lceil \log_2 N \rceil$ rounds in total. For notational convenience, let us assume that $N$ is a power of 2 for the rest of this analysis. At the

---

[2]In our notations, the zeroth level of the tournament contains $N$ initial candidates, the first level contains $N/2$ winners after the first round of pairwise comparisons, and so on. All candidates within the same level of the knockout tournament have the same probability of being a correct one, due to their symmetric roles.

$i$-th round, there are $N/2^i$ pairs of candidates, and each pair need $K$ comparisons; thus a total of $K \times N/2^i$ LLM calls are needed, which again can be parallelized.

In sum, the total number of LLM calls required by the two-stage algorithm is $N + K \times \sum_i N/2^i \leq (K+1) \times N$, whereas the end-to-end latency, if sufficiently many machines are available, is merely $T_{\text{gen}} + \log_2(N) \times T_{\text{comp}}$, where $T_{\text{gen}}$ and $T_{\text{comp}}$ represent the latency of one LLM call for generating a candidate solution and for comparing a pair of solutions, respectively.

## 3 A two-stage league-style algorithm

In this section, we propose a two-stage league-style algorithm that also enjoys a provable inference scaling law, under technical assumptions that are analogous to but more robust than those required by the knockout-style algorithm.

**The proposed algorithm.** To begin with, we generate $N$ candidate solutions $y_1, \ldots, y_N \sim \mathcal{M}_{\text{gen}}(x)$ as before.

---

**Algorithm 1** The proposed league-style algorithm

**Input:** the problem $x$.
1. Generate $N$ candidates $y_1, \ldots, y_N \sim \mathcal{M}_{\text{gen}}(x)$.
2. Compare each candidate $y_i$ against $K$ random opponents and estimate its average win rate $\widehat{\mu}_i$ by Eq. (2).
**Output:** the candidate with index $\widehat{i} := \arg\max_i \widehat{\mu}_i$.

---

Then, for each candidate with index $i \in [N]$, we randomly sample $K$ opponents with indices $o_i(1), \ldots, o_i(K) \in [N] \backslash \{i\}$ uniformly and with replacement, conduct one independent pairwise comparison against each opponent, and obtain the responses $r_i(j) \sim \mathcal{M}_{\text{comp}}(x, y_i, y_{o_i(j)})$ for $j \in [K]$. The *average win rate* of each candidate $y_i$ is then estimated by

$$\widehat{\mu}_i := \frac{1}{K} \sum_{j \in [K]} \phi\big(r_i(j), y_i, y_{o_i(j)}\big), \tag{2}$$

where $\phi\big(r_i(j), y_i, y_{o_i(j)}\big)$ denotes the score assigned, based on $r_i(j)$, to the candidate $y_i$ in its comparison against $y_{o_i(j)}$, e.g., 1 for a win, 0 for a loss, and 0.5 for a tie. Finally, the candidate with the highest average win rate $\widehat{\mu}_i$ is chosen (with ties broken randomly) as the output of the algorithm. See Algorithm 1 for a summary of this method.

Regarding computational efficiency, the proposed algorithm requires $N$ fully parallelizable LLM calls for the generation stage, and $N \times K$ fully parallelizable LLM calls for the aggregation stage.

**Analysis of success probability.** For a solution $y$, we denote its average win rate against $\mathcal{M}_{\text{gen}}$ by

$$\mu_y := \mathbb{E}_{y' \sim \mathcal{M}_{\text{gen}}(x)} \mathbb{E}_{r \sim \mathcal{M}_{\text{comp}}(x,y,y')} \phi(r, y, y').$$

Our key assumption is presented below.

**Assumption 3.1.** For the input problem $x$, there exist $p_{\text{cs}} > 0$, $\Delta > 0$, and a way of dividing the set $\mathcal{Y}$ of all possible solutions into three disjoint subsets $\mathcal{Y} = \mathcal{Y}_{\text{cs}} \cup \mathcal{Y}_{\text{cw}} \cup \mathcal{Y}_{\text{inc}}$ (where "cs", "cw" and "inc" stand for "correct-and-strong", "correct-but-weak" and "incorrect", respectively), such that

$$\mathbb{P}_{y \sim \mathcal{M}_{\text{gen}}(x)}(y \in \mathcal{Y}_{\text{cs}}) \geq p_{\text{cs}} > 0 \quad \text{and} \quad \min_{y \in \mathcal{Y}_{\text{cs}}} \mu_y - \max_{y \in \mathcal{Y}_{\text{inc}}} \mu_y \geq \Delta > 0.$$

In other words, we assume that the LLM can generate, with non-zero probability, a correct solution whose average win rate against $\mathcal{M}_{\text{gen}}$ is higher than that of any incorrect solution; such a solution is called correct-and-strong by our definition. We also allow the existence of correct-but-weak solutions, imposing no assumption on their average win rates. Note that Assumption 3.1 can be tolerant of systematic errors by LLMs in comparing certain pairs of candidates, i.e., it may still hold true when there exist a correct solution $y$ and incorrect solution $y'$ such that $\mathbb{E}_{r \sim \mathcal{M}_{\text{comp}}(x,y,y')} \phi(r, y, y') < 0.5$, whereas Assumption 2.1 fails in such cases.

*Remark* 3.2. One limitation of Assumption 3.1 is that it can be broken by an adversarial incorrect solution whose average win rate is unusually high, similar to the failure mode of best-of-N sampling discussed in Section 1. Nonetheless, on the presumption (backed by common practice and extensive empirical evidence) that pairwise comparison is more accurate and reliable than individual point-wise verification, we might safely say that Assumption 3.1 is conceptually weaker and more robust than the condition required by BoN (e.g., a perfect point-wise verifier) for provable inference scaling laws.

Intuitively, if Assumption 3.1 holds true and the hyperparameters $N$ and $K$ are sufficiently large, then with high probability, (1) there exist initial candidates that are correct-and-strong solutions, and (2) $\widehat{\mu}_i$ is an accurate estimate of $\mu_{y_i}$ for each $i \in [N]$. These conditions together lead to a correct final output of the algorithm. We formalize this intuition in the following theorem.

**Theorem 3.3.** *If Assumption 3.1 holds for the input problem, then the probability that the league-style algorithm (with hyperparameters $N$ and $K$) returns an incorrect final output is bounded by*

$$\mathbb{P}(\textit{failure}) \leq (1 - p_{cs})^N + 2Ne^{-K\Delta^2/8} + 2Ne^{-(N-1)\Delta^2/8}.$$

This theorem, whose proof can be found in Appendix B.3, ensures that the failure probability of the league-style algorithm decays to zero *exponentially* with respect to $N$ and $K$. Another way to interpret this theorem is as follows: to guarantee success probability $1 - \delta$, it suffices to have

$$N \geq \max\left\{\frac{1}{p_{cs}}\log\left(\frac{3}{\delta}\right), \frac{8}{\Delta^2}\log\left(\frac{6N}{\delta}\right) + 1\right\} \quad \text{and} \quad K \geq \frac{8}{\Delta^2}\log\left(\frac{6N}{\delta}\right).$$

That is, $N$ and $K$ have logarithmic dependence on $1/\delta$, and linear dependence on $\max\{1/p_{cs}, 1/\Delta^2\}$ and $1/\Delta^2$ (up to logarithmic factors) respectively.

*Remark* 3.4. The provable success of the knockout-style algorithm relies on Assumption 2.1, while that of the league-style algorithm relies on Assumption 3.1. Although the latter is conceptually more robust than the former, we note that neither assumption is strictly weaker than the other (and thus both algorithms have their unique values). In other words, there exist scenarios where Assumption 2.1 holds true while Assumption 3.1 does not, and also scenarios where the reverse is true. Interested readers may refer to Appendix C for some minimal examples.

## 4 Experiments

We conduct empirical studies to validate the efficacy and scaling properties of the proposed algorithms, while bridging their practical performance with the theories developed in previous sections.

**Datasets.** We use three datasets for our experiments: GPQA [33], MMLU-Pro [42] and MATH-500 [26]. GPQA consists of over 1000 graduate-level multiple-choice questions splitted into three categories ("main", "diamond" and "extended"), all of which are used in our experiments. MMLU-Pro contains 14 categories of multiple-choice questions, some of which require advanced reasoning while others are more knowledge-focused. Due to limited computational resources, we use a randomly sampled subset of 100 questions for each category of MMLU-Pro in our experiments, which leads to a total of 1400 questions; we refer to this subset as MMLU-Pro-S throughout this work. MATH-500 is a subset of 500 problems from the MATH dataset introduced in [22]. Due to space limitations, we focus mainly on GPQA in this section, deferring empirical results for MMLU-Pro-S and MATH-500 (as well as supplementary results for GPQA) to Appendix D.

**Implementations.** We use Llama3.1-70B-Instruct (`Llama3.1` for short) [24] and Qwen2.5-72B-Instruct (`Qwen2.5` for short) [49] in our experiments, as well as a `Mixed` option that uses a mixture of both LLMs [39, 54, 13]: during the generation stage, half of the initial candidates are sampled by `Llama3.1` and the other half by `Qwen2.5`; similarly, when a pair of candidates are compared for multiple times during the aggregation stage, half of them are done by `Llama3.1` and the other half by `Qwen2.5`. The rationale is that the capabilities of different LLMs can be complementary to some extent, and thus using a mixture of them can make it more likely that Assumptions 2.1 and 3.1 hold true[3]. Other models considered in our experiments include `QwQ-32B` [31], a long-CoT reasoning LLM, and `GPT-4o` [10], a proprietary API-based LLM; due to high computational or monetary costs, they are tested only on GPQA-diamond for a smaller range of $N$.

---

[3]To formalize this intuition, consider a minimal scenario with two LLMs denoted by $M_1$ and $M_2$, and two problems denoted by $x_1$ and $x_2$. Suppose that $M_1$ is effective for the first problem $x_1$ (with $p_{gen} = 0.2$ and $p_{comp} = 0.7$) but ineffective for $x_2$ (with $p_{gen} = 0$ and $p_{comp} = 0.5$), while the reverse holds true for $M_2$. When either LLM is used alone, only one problem satisfies Assumption 2.1. However, when a mixture of two LLMs is used, both problems now satisfy Assumption 2.1 with $p_{gen} = (0 + 0.2)/2 = 0.1 > 0$ and $p_{comp} = (0.5 + 0.7)/2 = 0.6 > 0.5$, and thus can be solved by our algorithm.

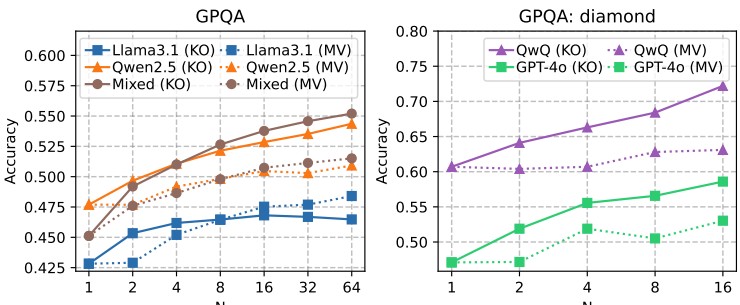

Figure 2: Accuracy versus the number of initial candidates $N$ for the knockout-style algorithm (KO), as well as for majority voting (MV), a strong baseline widely adopted in practice.

We leverage zero-shot chain-of-thought prompting [18] for both generation and aggregation stages of the proposed algorithms. Unless specified otherwise, for the knockout-style algorithm, we fix $K = 4$ for `Llama3.1`/ `Qwen2.5`/ `Mixed`, and $K = 2$ for `GPT-4o`/ `QwQ-32B`; for the league-style algorithm, we consider a round-robin [46] version of it, with $K = 4$ comparisons conducted between each of $\binom{N}{2}$ pairs of initial candidates. To make the proposed algorithms efficient and scalable in practice, we implement them based on AgentScope [9], a multi-agent framework that supports parallel and distributed computation[4]. Further implementation details can be found in Appendix D.1.

In our experiments, we consider a solution as a correct one if its final answer matches the ground-truth answer, and use accuracy (i.e., the proportion of correctly solved problems) as the performance metric for running a (deterministic or randomized) algorithm once on a dataset. This metric is, in expectation, equivalent to the mean success probability of the algorithm on the dataset.

## 4.1 Results for the knockout-style algorithm

**Efficacy and scaling properties.** Figure 2 confirms that the accuracy of the knockout-style algorithm improves with $N$ for all LLMs on GPQA or GPQA-diamond. For example, the accuracy of `Mixed` improves by 10 points (from $45\%$ to $55\%$) as $N$ scales to 64, and the accuracy of `QwQ-32B` improves by 12 points (from $60\%$ to $72\%$) as $N$ scales to 16. We also observe that `Mixed` consistently outperforms `Llama3.1` and `Qwen2.5` as $N$ gets larger, which confirms the previously explained rationales for using a mixture of different LLMs.

**Comparison with majority voting.** Figure 2 includes results for majority voting, a strong baseline widely adopted in practice. It is observed that, for all LLM backends (except for `Llama3.1`), the knockout-style algorithm consistently achieves higher accuracy when given the same number $N$ of initial candidates. Caution should be taken here: recall from Section 2.2 that the knockout-style algorithm takes $(K + 1) \times N$ LLM calls for solving one problem, i.e., $5 \times N$ for `Llama3.1`/ `Qwen2.5`/ `Mixed` and $3 \times N$ for `GPT-4o`/ `QwQ-32B`, whereas majority voting only requires $N$ LLM calls. Nonetheless, the knockout-style algorithm still has advantage when this is taken into account, e.g., its accuracy at $N = 8$ (resp. 4) is higher than that of majority voting at $N = 64$ (resp. 16) for `Mixed` (resp. `QwQ-32B`). Moreover, based on the trends shown in Figure 2, it is most likely that for majority voting, further increasing $N$ will bring limited performance gains [3] and result in a converged accuracy lower than what can be achieved by the knockout-style algorithm.

**But the theorems promise 100% accuracy, don't they?** The results in Figure 2 are indeed consistent with the theorems developed in Section 2.1, which guarantee that the knockout-style algorithm can achieve an arbitrarily high success probability for any input problem *satisfying Assumption 2.1*, namely $p_{\text{gen}} > 0$ and $p_{\text{comp}} > 0.5$. For a problem that does not, it is still possible that the algorithm has a chance of solving it correctly (since Assumption 2.1 is a sufficient condition for its success and might not be necessary), but there is no formal guarantee. Consequently, for a benchmark or a distribution of input problems, denoted by $\mathcal{D}$, our algorithm is guaranteed to achieve accuracy at least $\mathbb{P}_{x \sim \mathcal{D}}(x \text{ satisfies the assumption})$ as its test-time compute grows. Indeed, if a benchmark contains an extremely difficult problem, e.g., "solve the P versus NP problem", then any test-time scaling method will fail to achieve 100% accuracy on such a benchmark.

---

[4]Our implementations can be found at `https://github.com/pan-x-c/AgentScope/tree/feature/pxc/paper_provable/examples/paper_provable_scaling_law`

To further bridge the empirical results with theories, let us start by estimating the parameters $p_{gen}$ and $p_{comp}$ in Assumption 2.1. For each problem, we define $\widehat{p}_{gen}$ as the proportion of the $N = 64$ initial candidate solutions with a correct final answer, which serves as a good proxy for $p_{gen}$. To find a proxy for $p_{comp}$, we define $\widehat{p}_{comp}$ by picking all LLM calls for comparing a pair of correct and incorrect solutions throughout the knockout tournament, putting higher weights on the comparison results from later rounds of the tournament, and taking the sum of the weights of comparisons that identify the right winners[5].

Figure 7 in Appendix D.2 characterizes the distribution of GPQA and MMLU-Pro-S problems in terms of $\widehat{p}_{gen}$ (the X-axis) and $\widehat{p}_{comp}$ (the Y-axis); one such plot can also be found in Figure 3 (left). On the top half of each scatter plot are problems with $\widehat{p}_{comp} > 0.5$, most of which are solved correctly by the knockout-style algorithm and represented as circles. These include some

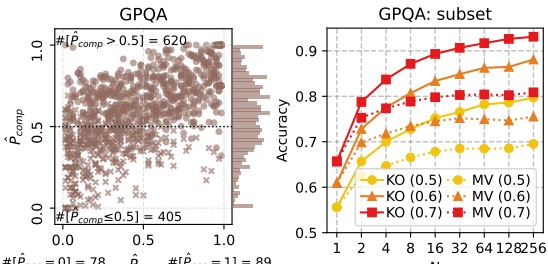

Figure 3: **Left:** the distribution of GPQA problems, characterized by $\widehat{p}_{gen}$ and $\widehat{p}_{comp}$ that are estimated with the knockout-style algorithm (`Mixed`). Each problem is represented by a circle if it is solved correctly at $N = 64$, and by a cross otherwise. **Right:** accuracy versus $N$ for the knockout-style algorithm and majority voting (both with `Mixed`), on a filtered subset of problems satisfying $0 < \widehat{p}_{gen} < 1$ and $\widehat{p}_{comp} > \tau$, where $\tau \in \{0.5, 0.6, 0.7\}$. The values of accuracy are calculated with new trials of the algorithm, thus statistically independent of $\widehat{p}_{gen}$ and $\widehat{p}_{comp}$.

problems with small $\widehat{p}_{gen}$, for which the knockout stage successfully identifies a correct candidate even though the initial candidates are mostly incorrect. We further observe from Figure 7 that, compared to `Llama3.1` and `Qwen2.5`, the `Mixed` option achieves $\widehat{p}_{gen} > 0$ and $\widehat{p}_{comp} > 0.5$ for a larger proportion of problems, which explains its superior accuracy shown in Figure 2. To further consolidate this analysis, we pay special attention to the subset of problems satisfying $0 < \widehat{p}_{gen} < 1$ and $\widehat{p}_{comp} > 0.5$. These are approximations for the conditions stated in Assumption 2.1, except that those easy problems with $\widehat{p}_{gen} = 1$ are excluded. We run new, independent trials of the knockout-style algorithm (`Mixed`) on this subset. Figure 3 (right) confirms that significant improvements in accuracy (from 55% to 80%) can be achieved by scaling up $N$, which matches what our theories predict. Unsurprisingly, the scaling curve still plateaus eventually (since $\widehat{p}_{comp} > 0.5$ is merely a proxy for $p_{comp} > 0.5$), and tightening the filtering condition (e.g., $\widehat{p}_{comp} > 0.6$ or 0.7) will bring it closer to 100% accuracy.

**Intuitions: when does Assumption 2.1 hold?** Interestingly, we observe that the scaling properties of the algorithm vary across different categories of MMLU-Pro-S, and also across LLM backends. For instance, Figure 4 shows that the performance scales well for all of `Llama3.1`/ `Qwen2.5`/ `Mixed` in the "engineering" category, while the scaling of `Llama3.1` outperforms the other two options in "philosophy". An intuitive explanation is that, for a reasoning-focused task like "engineering", LLMs can compare the reasoning processes of two candidate solutions side by side, which provides additional infor-

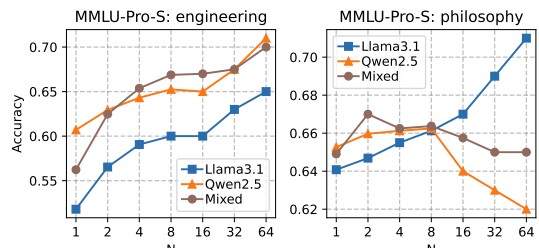

Figure 4: Accuracy versus $N$ for the knockout-style algorithm on two categories of MMLU-Pro-S.

mation compared to generating or verifying an individual solution, and thus leads to a large value of $p_{comp}$ and accurate comparison results. In contrast, for a knowledge-heavy task like "philosophy",

---

[5]The rationale for weighting the comparison results is explained as follows. In the early rounds of the knockout tournament, the comparison result between a correct-but-weak candidate and an incorrect candidate can cause a negative bias in estimating $p_{comp}$; similarly, a correct candidate might have a very high win rate against an opponent that is not only incorrect but also very weak, which can cause a positive bias in estimating $p_{comp}$. In contrast, the correct or incorrect candidates that survive the early rounds of the knockout tournament tend to be stronger ones, which make the comparison results among them (in later rounds of the tournament) more reliable and meaningful for the purpose of estimating $p_{comp}$.

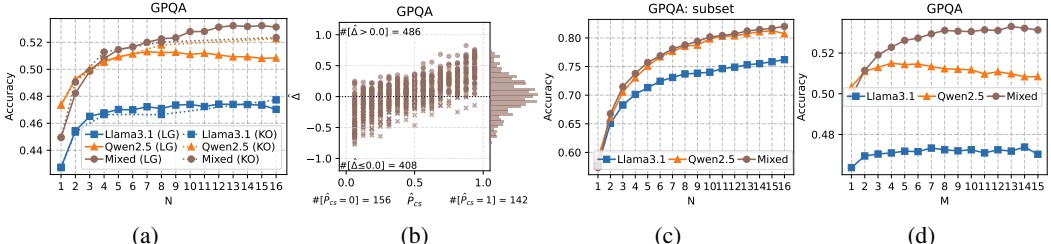

Figure 5: Empirical results for the league-style algorithm. **(a)** Accuracy versus the number of initial candidates $N$ for the league-style (LG, solid lines) and knockout-style (KO, dotted lines) algorithms, given the same initial candidates. **(b)** The distribution of GPQA problems, characterized by $\widehat{p}_{cs}$ and $\widehat{\Delta}$ that are estimated with the Mixed option. Each problem is represented by a circle if it is solved correctly at $N = 16$, and by a cross otherwise. **(c)** Accuracy versus $N$ on the subset of problems satisfying $0 < \widehat{p}_{cs} < 1$ and $\widehat{\Delta} > 0$. The values of accuracy are calculated with new trials of the algorithm, thus statistically independent of $\widehat{p}_{cs}$ and $\widehat{\Delta}$. **(d)** Accuracy versus $M$, the number of subsampled opponents for each candidate, for the league-style algorithm with $N = 16$.

one would not expect significant gains from pairwise comparison if the LLM simply does not have the right knowledge embedded within its model weights, in which case $p_{comp}$ might be close to (or even below) $0.5$.

## 4.2 Results for the league-style algorithm

**Efficacy and scaling properties.** Figure 5 (a) shows that the accuracy of the league-style algorithm grows with $N$ for all of Llama3.1, Qwen2.5 and Mixed options on GPQA, e.g., accuracy of Mixed improves by 8 points (from $45\%$ to $53\%$) as $N$ grows to 16. The Mixed option consistently outperforms Llama3.1 and Qwen2.5, similar to the case for the knockout-style algorithm. Given the same initial candidates, the league-style algorithm achieves higher accuracy than the knockout-style algorithm does in some cases and lower in other cases, although the differences are minor in general.

**Bridging with the theories.** Let us start by finding proxies for the $p_{cs}$ and $\Delta$ parameters in Assumption 3.1. For the former, we define $\widehat{p}_{cs}$ as the fraction of initial candidate solutions with the correct final answer. For the latter, we define $\widehat{\Delta} := \widehat{\mu}_{i_1} - \widehat{\mu}_{i_2}$, where $i_1 := \arg\max_{i \in [N]: y_i \text{ is correct}} \widehat{\mu}_i$ and $i_2 := \arg\max_{i \in [N]: y_i \text{ is incorrect}} \widehat{\mu}_i$ are the indices for the strongest correct candidate and for the strongest incorrect candidate, respectively. Note that by definition, for any problem with $\widehat{p}_{cs} \notin \{0, 1\}$, the league-style algorithm returns a correct solution to the problem if and only if $\widehat{\Delta} > 0$.

Figure 15 in Appendix D.3 characterizes the distribution of GPQA and MMLU-Pro-S problems in terms of $\widehat{p}_{cs}$ (X-axis) and $\widehat{\Delta}$ (Y-axis); the plot corresponding to GPQA and Mixed can also be found in Figure 5 (b). It is noteworthy that there exists a non-trivial proportion of problems for which $\widehat{p}_{cs}$ is fairly small (i.e., most of the initial candidates are incorrect), yet the proposed league-style aggregation stage still manages to attain a positive $\widehat{\Delta}$ and thus identify a correct candidate for the final output. On the other hand, for problems with $\widehat{\Delta} < 0$ (which indicates violations of Assumption 3.1), there is no success guarantee for the algorithm. Figure 5 (c) further confirms that significant improvements in accuracy, e.g., a $25\%$ increase for Mixed, can be achieved on the subset of problems that approximately satisfy Assumption 3.1.

**Efficacy of subsampling opponents.** While all previous experiments consider the round-robin version of the league-style algorithm, we also wonder if it is feasible to improve its computational efficiency by estimating the average win rate of each candidate with $M < N - 1$ subsampled opponents, while maintaining its accuracy. The empirical results in Figure 5 (d) provide a positive answer and match what our theories in Section 3 predict: (1) accuracy initially increases with $M$, which confirms the benefits of comparing each candidate with multiple opponents; (2) once $M$ exceeds a threshold (around 4 or 5) that is much smaller than $N = 16$, accuracy saturates around the level achieved by the round-robin version, but at a lower computational cost.

# 5 Related works

There exist other test-time strategies that enjoy provable inference scaling laws in the sense of Definition 1.1. One example is majority voting, whose provable success requires two assumptions [3, 47]: (1) it is feasible to divide the candidate solutions into several groups and have a meaningful count for each group (which is not the case in tasks like open-ended writing, where all candidate solutions are distinct), and (2) the probability that one LLM call generates a solution belonging to the correct group is higher than that for any other group. In comparison, our proposed algorithms are free from the first restriction, and only require $p_{\text{gen}} > 0$ while making additional assumptions about LLMs' capabilities in pairwise comparisons. Another example is best-of-N (BoN) sampling, for which deriving a provable scaling law is straightforward *provided that* a perfect verifier is available: if one LLM call generates a correct solution with probability $p_{\text{gen}} > 0$, then the failure probability of BoN is $(1 - p_{\text{gen}})^N$. One obvious limitation is that verifiers are unavailable or imperfect in many practical scenarios, which can hinder the performance of BoN [5, 36, 2]. We refrain from comparing our methods with BoN in our experiments, since introducing an external verifier or reward model will bring extra variability that makes it difficult to conduct a fair and meaningful empirical comparison.

Our algorithm design has drawn inspiration from various areas. For example, the essential idea of pairwise comparison has been prominent in LLM alignment [1, 30, 32] and the LLM-as-a-judge paradigm [57, 20]. Although it is possible to verify, score or refine a solution by itself [14, 25, 6, 11], it is often much easier (for LLMs or human) to detect the errors or hallucinations in an incorrect solution when it is placed right next to a correct one, or evaluate the quality of a solution by comparing it to another one. The knockout and league tournaments have also been investigated in prior LLM research [17, 23, 56, 19, 13], albeit with purposes or implementations that are different from ours. Given this context, we remark that the main novelty and contributions in this work are perhaps less about the proposed two-stage algorithms themselves, but rather more about developing rigorous and theoretical understanding of their underlying assumptions and efficacy (via clearly identifying sufficient conditions for boosting their success probability up to $100\%$ and formally deriving quantitative bounds for their computational and sample complexities), and demonstrating their promising empirical performance through extensive experiments.

# 6 Limitations and future work

One limitation of this work is that, like any other test-time scaling method, the proposed algorithms trade computation for a higher success rate. Future work may try to find practical ways to determine the smallest values of hyperparameters $N$ and $K$ necessary for a targeted success probability. We also note that the provable success of the proposed algorithms relies on technical assumptions that might not always hold true (as is the case for many other theories), although we anticipate optimistically that with the ongoing developments of LLMs, the assumptions made in this work (and thus our algorithms) will *automatically* become feasible for more and more challenging tasks.

Future work may also try to extend the methodologies and theories to broader scenarios, including (1) evaluating the proposed algorithms in more diverse tasks; (2) combining the proposed algorithms with other test-time scaling strategies for the best performance [35]; (3) efficiently amplifying the success probability of an agentic workflow by applying the proposed algorithms to each sub-task; (4) converting the proposed methods to anytime algorithms [45] in online scenarios where the amount of available test-time compute is adaptive and unknown a priori. We defer detailed discussions to Appendix A, due to space limitations.

# Acknowledgments

The authors would like to thank the anonymous reviewers and Area Chairs for their constructive feedback that has helped improve this work.

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

## Structure of the appendix

This appendix is organized as follows. Appendix A supplements the discussions in Section 6 about potential extensions of the current work to broader scenarios. Appendix B includes the proofs for our main theorems, and Appendix C presents some minimal examples that assist our understanding about how Assumptions 2.1 and 3.1 compare with each other. Appendix D includes additional implementation details and the prompt templates used throughout our experiments, as well as supplementary empirical results.

## A  Discussions: extensions to broader scenarios

**Combination with other test-time strategies.**   The proposed two-stage methods are orthogonal and complementary to many other test-time scaling strategies. For example, our experiment with `QwQ-32B` shows that the performance of a long-CoT reasoning LLM can be further boosted with the knockout-style algorithm. Future work may investigate systematic approaches of combining different inference scaling strategies for achieving the best performance [35].

**Application to agentic workflows.**   In complex real-world scenarios, a common practice is to adopt an agentic workflow that decomposes the original task into manageable sub-problems and involves multiple LLM calls to solve all of them [52, 48, 4]. Applying the knockout/league-style algorithm proposed in this work to each sub-problem can efficiently amplify the success probability of the overall workflow. To see how this works, consider a scenario where solving the original problem requires solving all $S \geq 1$ sub-problems correctly, and each sub-problem satisfies Assumption 2.1 with parameters $p_{\text{gen}} > 0$ and $p_{\text{comp}} > 0.5$. Directly solving all $S$ sub-problems has a *exponentially small* success probability $p_{\text{gen}}^S$, and thus generating a correct solution alone already requires $\Omega((1/p_{\text{gen}})^S)$ attempts, not to mention identifying which attempt is successful. In contrast, by applying the knockout-style algorithm (with hyperparameters $N$ and $K$) to each sub-problem, an overall success probability $1 - \delta$ for solving the original problem can be guaranteed as long as the failure probability for each sub-problem is bounded by $\delta/S$, thanks to the union bound. According to Eq. (1), this is guaranteed with

$$N \geq \frac{1}{p_{\text{gen}}} \log\left(\frac{2S}{\delta}\right) \quad \text{and} \quad K \geq \frac{1}{2(p_{\text{comp}} - 0.5)^2} \log\left(\frac{2\lceil \log_2 N \rceil S}{\delta}\right).$$

both with logarithmic dependence on $S$. The total number of LLM calls with this approach is $(K + 1) \times N \times S$ (cf. Section 2.2), which grows with $S$ *linearly*, up to logarithmic factors.

**Anytime algorithms for online settings.**   In many real-world scenarios, the available amount of test-time compute is adaptive and unknown a priori. To address such cases, we can easily convert the knockout-style algorithm to an "anytime" variant [45] that does not require pre-specifying $N$. For example, the algorithm might start with 4 candidate solutions and choose the winner via a knockout tournament. If more test-time compute is allowed (e.g., the user is not eagerly requesting the solution, or more computational resources become available), then the algorithm can launch another tournament with 4 freshly sampled candidates, the winner of which will compete with the winner of the previous tournament. This complete process is indeed equivalent to a single tournament with $N = 4 + 4 = 8$. Such a process can continue until the user finally requests the solution; the eventual value of $N$ is determined online and automatically achieves the maximum value allowed by the available test-time compute. Similarly, the league-style algorithm can be converted to an anytime variant, where the total number of candidates and/or the number of comparisons for each candidate increase gradually as more test-time compute becomes available. It would be interesting future work to investigate such anytime algorithms from a theoretical or practical perspective.

## B  Proofs of main theorems

### B.1  Proof of Theorem 2.3

To begin with, we have a straightforward analysis for the failure probability of the generation stage of the algorithm, where $N$ candidate solutions are sampled independently:

$$\mathbb{P}(\text{no candidate solution is correct}) \leq (1 - p_{\text{gen}})^N.$$

As for the knockout stage, let us first consider a single pair of correct and incorrect candidate solutions. Recall that they are compared for $K$ times with $K$ LLM calls (followed by majority voting), and each LLM call identifies the correct candidate solution as the winner with probability $\mu \geq p_{\text{comp}} > 0.5$ by assumption. Therefore, the failure probability of comparing this pair of candidates can be bounded as follows, where $X_i$ denotes an independent Bernoulli random variable with mean $\mu$:

$$\mathbb{P}(\text{failure of comparison}) \leq \mathbb{P}\Big( \sum_{i \in [K]} X_i \leq \frac{K}{2} \Big) = \mathbb{P}\Big( \frac{1}{K} \sum_{i \in [K]} X_i \leq 0.5 \Big)$$

$$= \mathbb{P}\Big( \frac{1}{K} \sum_{i \in [K]} X_i - \mu \leq -(\mu - 0.5) \Big)$$

$$\leq \exp\Big( -2K(\mu - 0.5)^2 \Big) \leq \exp\Big( -2K(p_{\text{comp}} - 0.5)^2 \Big).$$

Here we use Hoeffding's inequality [38] in the last line.

Now we are ready to control the failure probability of the complete knockout stage. Let us condition on the event that the generation stage succeeds, i.e., there is at least one initial candidate solution that is correct. We arbitrarily pick a correct candidate, and focus on its path to the final output of the algorithm in the binary tree visualized in Figure 1. We claim that, with high probability, the comparison (with $K$ LLM calls) for each pair along this path yields the correct outcome. This can be proved by induction: for each pair along this path, if one of the input candidates (which is the output of the previous pairwise comparison on this same path) is correct, then the output of comparing this pair will also be correct with a failure probability no greater than $\exp(-2K(p_{\text{comp}} - 0.5)^2)$, regardless of whether the other input candidate is correct or not. By taking a union bound over the failure events along this path with $\lceil \log_2 N \rceil$ pairs to be compared, we claim that the comparison for each pair along this path yields the correct outcome (which immediately implies that the final output of the algorithm is correct), with a failure probability no greater than $\lceil \log_2 N \rceil \exp(-2K(p_{\text{comp}} - 0.5)^2)$.

Finally, taking a union bound over the failure events of both stages of the algorithm completes our proof of Theorem 2.3.

*Remark* B.1. There exists analysis in the literature of top-k ranking (e.g., Section 4.1 of [27]) that are similar to our analysis for the knockout stage. We choose to present our own version here to make our work more self-contained and complete.

## B.2 Proof of Theorem 2.4

For the first part of the theorem, we can derive $p_{i+1}$ from $p_i$ as follows. Notice that a candidate at the $(i + 1)$-th level of the knockout tournament is the winner of pairwise comparison between a pair of *statistically independent* candidates at the $i$-th level. Thus, the winner is a correct solution if both candidates of the pair are correct, or only one of them is correct and happens (with probability at least $p_{\text{comp},K}$ by assumption) to be chosen as the winner. Therefore,

$$p_{i+1} \geq p_i^2 + 2p_i(1 - p_i)p_{\text{comp},K} = p_i^2 + 2p_{\text{comp},K}(p_i - p_i^2) = p_i - p_i + p_i^2 + 2p_{\text{comp},K}(p_i - p_i^2)$$

$$= p_i + (2p_{\text{comp},K} - 1)(p_i - p_i^2).$$

This implies $p_{i+1} > p_i$, as long as $p_{\text{comp},K} > 0.5$ and $p_i < 1$.

For the second part of the theorem, we consider the convergence of $\{p_i\}$ in two cases: when it is still below $0.5$, and when it has exceeded $0.5$.

- If $p_i < 0.5$, then $1 - p_i > 0.5$, and

$$p_{i+1} \geq p_i + (2p_{\text{comp},K} - 1)(1 - p_i)p_i$$

$$> p_i + (p_{\text{comp},K} - 0.5)p_i$$

$$= \Big( 1 + (p_{\text{comp},K} - 0.5) \Big)p_i.$$

In other words, the sequence $\{p_i\}$ grows exponentially when it is below $0.5$, and

$$p_J \geq p_{\text{gen}}\Big( 1 + (p_{\text{comp},K} - 0.5) \Big)^J \geq 0.5 \quad \text{as long as} \quad J \geq \frac{\log\big( \max\{\frac{1}{2p_{\text{gen}}}, 1\}\big)}{\log\big(1 + (p_{\text{comp},K} - 0.5)\big)}.$$

$$(3)$$

- For any $i > J$ and hence $p_i \geq 0.5$, we have

$$1 - p_{i+1} \leq 1 - p_i - (2p_{\text{comp},K} - 1)p_i(1 - p_i)$$
$$\leq 1 - p_i - (p_{\text{comp},K} - 0.5)(1 - p_i)$$
$$= \Big(1 - (p_{\text{comp},K} - 0.5)\Big)(1 - p_i).$$

In other words, the sequence $\{1 - p_i\}$ converges to 0, and

$$1 - p_i \leq (1 - p_J)\Big(1 - (p_{\text{comp},K} - 0.5)\Big)^{i-J} \leq \frac{1}{2}\Big(1 - (p_{\text{comp},K} - 0.5)\Big)^{i-J} \leq \delta$$

$$\text{as long as} \quad i - J \geq \frac{\log\left(\frac{1}{2\delta}\right)}{-\log\left(1 - (p_{\text{comp},K} - 0.5)\right)}. \tag{4}$$

Putting Eq. (3) and Eq. (4) together concludes our proof of the theorem.

### B.3 Proof of Theorem 3.3

To begin with, we have a straightforward analysis for the generation stage:

$$\mathbb{P}\Big(y_i \notin \mathcal{Y}_{\text{cs}}, \forall i \in [N]\Big) \leq (1 - p_{\text{cs}})^N.$$

For the aggregation stage, we aim to show that for each $i \in [N]$, the estimated average win rate $\widehat{\mu}_i$ calculated within the algorithm is close to its average win rate against $\mathcal{M}_{\text{gen}}$, denoted by $\mu_i := \mu_{y_i}$. To see this, let us recall the definitions of $\mu_i$ and $\widehat{\mu}_i$, as well as introduce a new notation $\widetilde{\mu}_i$:

$$\mu_i := \mathbb{E}_{y' \sim \mathcal{M}_{\text{gen}}(x)} \mathbb{E}_{r \sim \mathcal{M}_{\text{comp}}(x, y_i, y')} \phi(r, y_i, y'),$$

$$\widetilde{\mu}_i := \frac{1}{N-1} \sum_{j \in [N] \setminus \{i\}} \mathbb{E}_{r \sim \mathcal{M}_{\text{comp}}(x, y_i, y_j)} \phi(r, y_i, y_j)$$

$$= \mathbb{E}_{y' \sim \text{Unif}(y_j, j \in [N] \setminus \{i\})} \mathbb{E}_{r \sim \mathcal{M}_{\text{comp}}(x, y_i, y')} \phi(r, y_i, y'),$$

$$\widehat{\mu}_i := \frac{1}{K} \sum_{j \in [K]} \phi(r_i(j), y_i, y_{o_i(j)}).$$

Note that in the last line, $y_{o_i(j)} \sim \text{Unif}(y_j, j \in [N] \setminus \{i\})$, and $r_i(j) \sim \mathcal{M}_{\text{comp}}(x, y_i, y_{o_i(j)})$. By Hoeffding's inequality, we have the following for each $i \in [N]$:

$$\mathbb{P}\Big(|\widehat{\mu}_i - \widetilde{\mu}_i| \geq \frac{\Delta}{4}\Big) = \mathbb{P}\Big(\Big|\frac{1}{K} \sum_{j \in [K]} \phi(r_i(j), y_i, y_{o_i(j)}) - \widetilde{\mu}_i\Big| \geq \frac{\Delta}{4}\Big) \leq 2\exp\Big(-\frac{K\Delta^2}{8}\Big),$$

$$\mathbb{P}\Big(|\widetilde{\mu}_i - \mu_i| \geq \frac{\Delta}{4}\Big) = \mathbb{P}\Big(\Big|\frac{1}{N-1} \sum_{j \in [N] \setminus \{i\}} \mathbb{E}_{r \sim \mathcal{M}_{\text{comp}}(x, y_i, y_j)} \phi(r, y_i, y_j) - \mu_i\Big| \geq \frac{\Delta}{4}\Big)$$

$$\leq 2\exp\Big(-\frac{(N-1)\Delta^2}{8}\Big).$$

These, together with the fact that $|\widehat{\mu}_i - \mu_i| \leq |\widehat{\mu}_i - \widetilde{\mu}_i| + |\widetilde{\mu}_i - \mu_i|$, implies that

$$\mathbb{P}\Big(|\widehat{\mu}_i - \mu_i| \geq \frac{\Delta}{2}\Big) \leq \mathbb{P}\Big(|\widehat{\mu}_i - \widetilde{\mu}_i| \geq \frac{\Delta}{4}\Big) + \mathbb{P}\Big(|\widetilde{\mu}_i - \mu_i| \geq \frac{\Delta}{4}\Big)$$

$$\leq 2\exp\Big(-\frac{K\Delta^2}{8}\Big) + 2\exp\Big(-\frac{(N-1)\Delta^2}{8}\Big).$$

Finally, taking a union bound over $i \in [N]$ and over both stages of the league-style algorithm, we have the following: with probability at least

$$1 - (1 - p_{\text{cs}})^N - 2N\exp\Big(-\frac{K\Delta^2}{8}\Big) - 2N\exp\Big(-\frac{(N-1)\Delta^2}{8}\Big),$$

there exists some $i \in [N]$ such that $y_i \in \mathcal{Y}_{\text{cs}}$, and $|\widehat{\mu}_j - \mu_j| < \Delta/2$ for all $j \in [N]$. These conditions, together with the assumption that $\min_{y \in \mathcal{Y}_{\text{cs}}} \mu_y - \max_{y \in \mathcal{Y}_{\text{inc}}} \mu_y \geq \Delta$, guarantee that the final output of the algorithm is a correct solution.

## C Examples for understanding and comparing the assumptions

This section presents some minimal examples for assisting our understanding of Assumptions 2.1 and 3.1, and in particular, for comparing the condition $p_{\text{comp}} > 0.5$ stated in the former and $\Delta > 0$ stated in the latter. For simplicity, we assume that the set of all possible candidate solutions returned by the generation stage, denoted by $\mathcal{Y}$, has a small number of unique elements, e.g., $\mathcal{Y} = \{A, B, C\}$. We use the notation $p_A := \mathbb{P}_{y \sim \mathcal{M}_{\text{gen}}(x)}(y = A)$, and let $\mathbb{P}(A \succ B)$ denote the probability that one comparison between $A$ and $B$ identifies the former as the winner. When two identical candidates are compared, we assume that tie is broken randomly and thus either candidate wins with probability 0.5. All average win rates involved in these examples are calculated with respect to the distribution $\mathcal{M}_{\text{gen}}$.

**Example C.1.** We demonstrate a scenario where both Assumptions 2.1 and 3.1 hold, and there is a correspondence between the parameter $p_{\text{comp}}$ in the former and $\Delta$ in the latter. Suppose that $\mathcal{Y} = \{A, B\}$, where $A$ is correct and $B$ is incorrect. In addition, $p_A = \alpha, p_B = 1 - \alpha$, and $\mathbb{P}(A \succ B) = p_{\text{comp}} > 0.5$. Then we can calculate the average win rate of each candidate as follows:

$$
\begin{aligned}
\mu_A &= p_A \times 0.5 + p_B \times p_{\text{comp}} = 0.5 \times \alpha + p_{\text{comp}} \times (1 - \alpha) \\
&= (0.5 - p_{\text{comp}}) \times \alpha + p_{\text{comp}}, \\
\mu_B &= p_B \times 0.5 + p_A \times (1 - p_{\text{comp}}) = 0.5 \times (1 - \alpha) + (1 - p_{\text{comp}}) \times \alpha \\
&= (0.5 - p_{\text{comp}}) \times \alpha + 0.5,
\end{aligned}
$$

which implies that

$$
\Delta = \mu_A - \mu_B = p_{\text{comp}} - 0.5 > 0
$$

is independent of the value of $\alpha$.

**Example C.2.** We demonstrate a scenario where both Assumptions 2.1 and 3.1 hold, but the parameter $\Delta$ in the latter can be much smaller than $p_{\text{comp}} - 0.5$ in the former. Suppose that $\mathcal{Y} = \{A, B, C\}$, where only $A$ is correct. In addition, $p_A = p_B = \alpha, p_C = 1 - 2\alpha$, $\mathbb{P}(A \succ B) = p_{\text{comp}} > 0.5$, and $\mathbb{P}(A \succ C) = \mathbb{P}(B \succ C) = 0.9$. Then we can calculate the average win rate of each candidate as follows:

$$
\begin{aligned}
\mu_A &= p_A \times 0.5 + p_B \times p_{\text{comp}} + p_C \times 0.9 &&= \alpha \times (0.5 + p_{\text{comp}}) + (1 - 2\alpha) \times 0.9, \\
\mu_B &= p_A \times (1 - p_{\text{comp}}) + p_B \times 0.5 + p_C \times 0.9 &&= \alpha \times (1.5 - p_{\text{comp}}) + (1 - 2\alpha) \times 0.9, \\
\mu_C &= p_A \times 0.1 + p_B \times 0.1 + p_C \times 0.5 &&= \alpha \times 0.2 + (1 - 2\alpha) \times 0.5,
\end{aligned}
$$

which implies

$$
\Delta = \mu_A - \mu_B = 2\alpha \times (p_{\text{comp}} - 0.5).
$$

As a result, $\Delta > 0$ can be much smaller than $p_{\text{comp}} - 0.5$ if $\alpha$ is small.

**Example C.3.** We demonstrate a scenario where Assumption 2.1 holds true but Assumption 3.1 does not. Suppose that $\mathcal{Y} = \{A, B, C\}$, where only $A$ is correct. In addition, $p_A = 0.2, p_B = 0.2, p_C = 0.6$, $\mathbb{P}(A \succ B) = \mathbb{P}(A \succ C) = 0.6$, and $\mathbb{P}(B \succ C) = 0.9$, which satisfies Assumption 2.1. Then we have

$$
\begin{aligned}
\mu_A &= p_A \times 0.5 + p_B \times 0.6 + p_C \times 0.6 = 0.58, \\
\mu_B &= p_A \times 0.4 + p_B \times 0.5 + p_C \times 0.9 = 0.72, \\
\mu_C &= p_A \times 0.4 + p_B \times 0.1 + p_C \times 0.5 = 0.40.
\end{aligned}
$$

In other words, the average win rate of the only correct solution $A$ is lower than that of an incorrect solution $B$, which violates Assumption 3.1.

**Example C.4.** We demonstrate a scenario where Assumption 3.1 holds true but Assumption 2.1 does not. Suppose that $\mathcal{Y} = \{A, B, C\}$, where only $A$ is correct. In addition, $p_A = 0.2, p_B = 0.2, p_C = 0.6$, $\mathbb{P}(A \succ B) = 0.4, \mathbb{P}(A \succ C) = 0.9$, and $\mathbb{P}(B \succ C) = 0.5$, which violates Assumption 2.1. However, we have

$$
\begin{aligned}
\mu_A &= p_A \times 0.5 + p_B \times 0.4 + p_C \times 0.9 = 0.72, \\
\mu_B &= p_A \times 0.6 + p_B \times 0.5 + p_C \times 0.5 = 0.52, \\
\mu_C &= p_A \times 0.1 + p_B \times 0.5 + p_C \times 0.5 = 0.42,
\end{aligned}
$$

which satisfies Assumption 3.1 since the only correct solution $A$ has the highest average win rate.

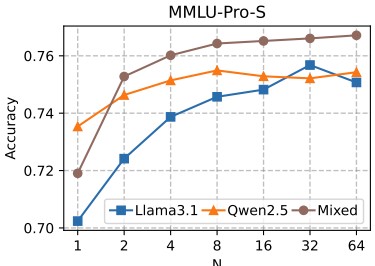
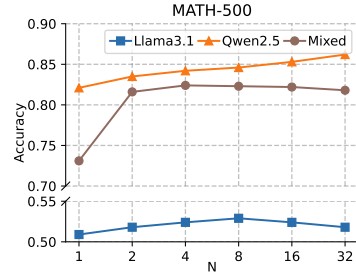

Figure 6: Accuracy versus the number of initial candidates $N$ for the knockout-style algorithm on MMLU-Pro-S (left) and MATH-500 (right).

## D Supplementary materials for experiments

### D.1 Additional implementation details

Throughout our experiments, the temperature for LLM decoding is set to 0.5 for the generation stage, and 0.1 for pairwise comparisons during the aggregation stage. Our early exploration suggests that these choices strike a good balance between diversity and preciseness in LLM decoding.

During the generation stage, we ask the LLM (via zero-shot CoT prompting [18]) to generate a reasoning process first and then its final answer. For each pairwise comparison during the aggregation stage, we also leverage zero-shot CoT prompting and ask the LLM to think step by step before deciding which solution in the pair is more plausible, unless specified otherwise. Tables 1 and 2 at the end of this section include the prompt templates used in our experiments for GPQA and MMLU-Pro-S, both of which are multiple-choice datasets. The prompt templates for MATH-500 are largely the same, only slightly adjusted to account for the desired output formats. Some parts of our prompts, as well as code for parsing LLMs' responses and extracting the answers for evaluation, are modified from those in the official GitHub repository of MMLU-Pro[6].

To account for the positional bias of LLMs [57, 40], we ensure that when a pair of candidates are compared for multiple times, they are placed in one order within the prompt for half of the comparisons, and in the opposite order for the other half.

Due to the high computational or monetary costs of the experiments, we have run the knockout/league-style algorithm only once for each <model, dataset> combination. To enhance the stability and reliability of the plots in this paper, we take the following approaches:

- For the knockout-style algorithm, we take advantage of its binary tree structure (shown in Figure 1). After running the algorithm once with $N = 64$, we automatically get the results of 64 independent trials for $N = 1$, 32 trials for $N = 2$, 16 trials for $N = 4$, and so on. We have thus taken the average of accuracy values from multiple independent trials for each datapoint (except for the rightmost one) in each scaling curve.

- For the league-style algorithm, after running it once with $N = 16$, we are able to obtain the results of multiple trials for $N = 8$ (or any value smaller than 16), each corresponding to 8 randomly sampled candidate solutions and the comparison results among them. Each datapoint (except for the rightmost one) in each scaling curve has been calculated by an average of multiple results obtained this way.

### D.2 Additional results for the knockout-style algorithm

**Experiments with more datasets.** Figure 6 validates the efficacy of the knockout-style algorithm on MMLU-Pro-S and MATH-500.

**Distribution of problems.** Figure 7 illustrates the distribution of GPQA and MMLU-Pro-S problems, characterized by $\widehat{p}_{\text{gen}}$ and $\widehat{p}_{\text{comp}}$ that are estimated with the empirical results for the knockout-

---

[6]https://github.com/TIGER-AI-Lab/MMLU-Pro/tree/main

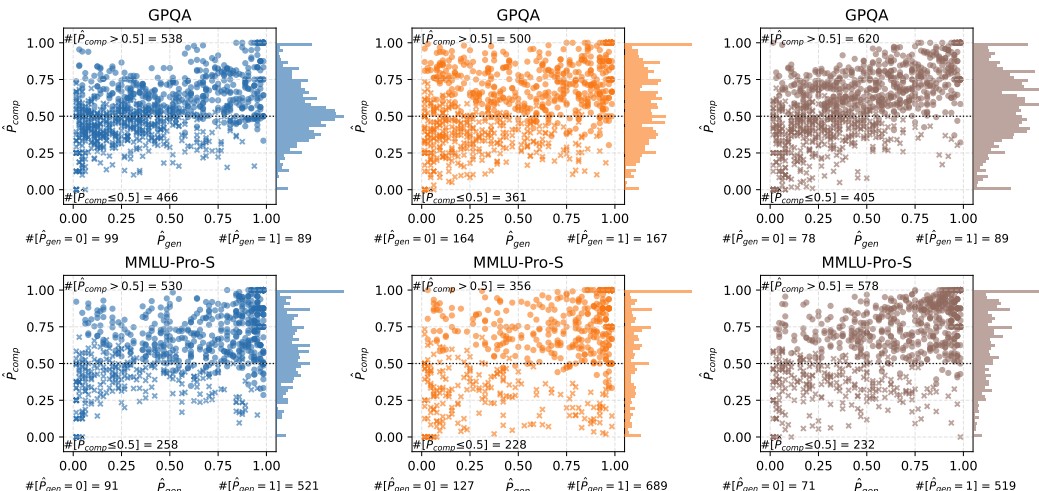

Figure 7: The distribution of GPQA (top) and MMLU-Pro-S (bottom) problems, characterized by $\widehat{p}_{gen}$ and $\widehat{p}_{comp}$ that are estimated with the empirical results for the knockout-style algorithm using the `Llama3.1` (left), `Qwen2.5` (middle) or `Mixed` (right) option. Each plot is annotated with the number of problems satisfying the condition $\widehat{p}_{comp} > 0.5$, $\widehat{p}_{comp} \leq 0.5$, $\widehat{p}_{gen} = 0$ or $\widehat{p}_{gen} = 1$. To the right of each plot is a histogram for $\widehat{p}_{comp}$. Each problem is represented by a circle if it is solved correctly by the knockout-style algorithm with $N = 64$, and by a cross otherwise. We neglect problems with $\widehat{p}_{gen} = 0$ or $1$, i.e., problems for which the initial candidate solutions are all incorrect or all correct, since there is no way of obtaining meaningful estimate of $\widehat{p}_{comp}$ for such problems.

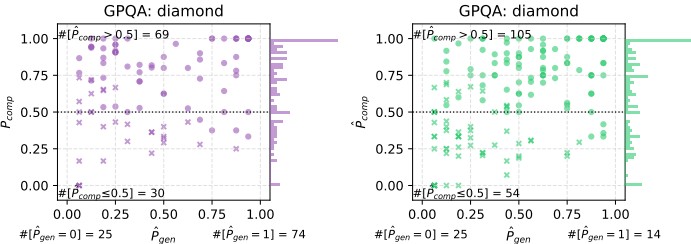

Figure 8: The distribution of GPQA-diamond problems, characterized by $\widehat{p}_{gen}$ and $\widehat{p}_{comp}$ that are estimated with the empirical results for the knockout-style algorithm using the `QwQ-32B` (left) or `GPT-4o` (right) option. Other settings are the same as those in Figure 7.

style algorithm using the `Llama3.1` (left), `Qwen2.5` (middle) or `Mixed` (right) option. Similarly, Figure 8 illustrates the results for `QwQ-32B` and `GPT-4o`.

**Ablation: the impact of $K$.** The results in Figure 9 suggest that the performance of the knockout-style algorithm is insensitive to $K$ (the number of times that each pair of candidates are compared) in the setting of our experiments, as long as $K \geq 2$ for `Llama3.1` and `Qwen2.5`, or $K \geq 4$ for `Mixed`. This is mainly due to our choice of a small temperature (0.1) for LLM calls that conduct pairwise comparisons. For `Llama3.1` and `Qwen2.5`, $K = 2$ suffices to cover all prompting options, i.e., the order in which two candidate solutions are placed within the prompt. Similarly, for `Mixed`, $K = 4$ suffices to cover both prompting options and both LLM backends.

**Ablation: the impact of CoT prompting for pairwise comparison.** Figure 10 confirms the benefits of using zero-shot chain-of-thought prompting for the aggregation stage of the knockout-style algorithm (versus prompting the LLM to answer directly which solution is preferred), especially as the test-time compute scales up. This matches the intuition that CoT prompting improves LLMs' performance in conducting pairwise comparisons.

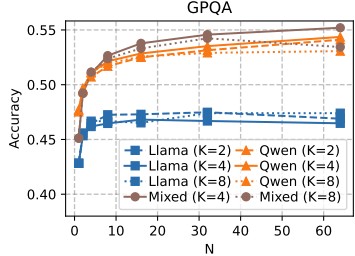

Figure 9: The impact of $K$ for the knockout-style algorithm.

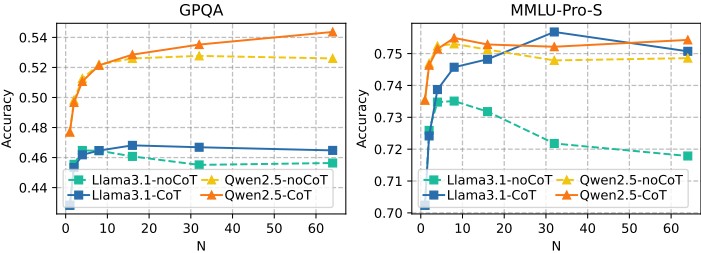

Figure 10: The advantages of zero-shot chain-of-thought prompting for pairwise comparisons, versus prompting the LLM to answer directly which solution is preferred (dashed lines), during the aggregation stage of the knockout-style algorithm.

**Results for each category of GPQA and MMLU-Pro-S.** Figure 11 includes empirical results of the knockout-style algorithm for each category of GPQA, while Figures 12 and 13 include those for MMLU-Pro-S.

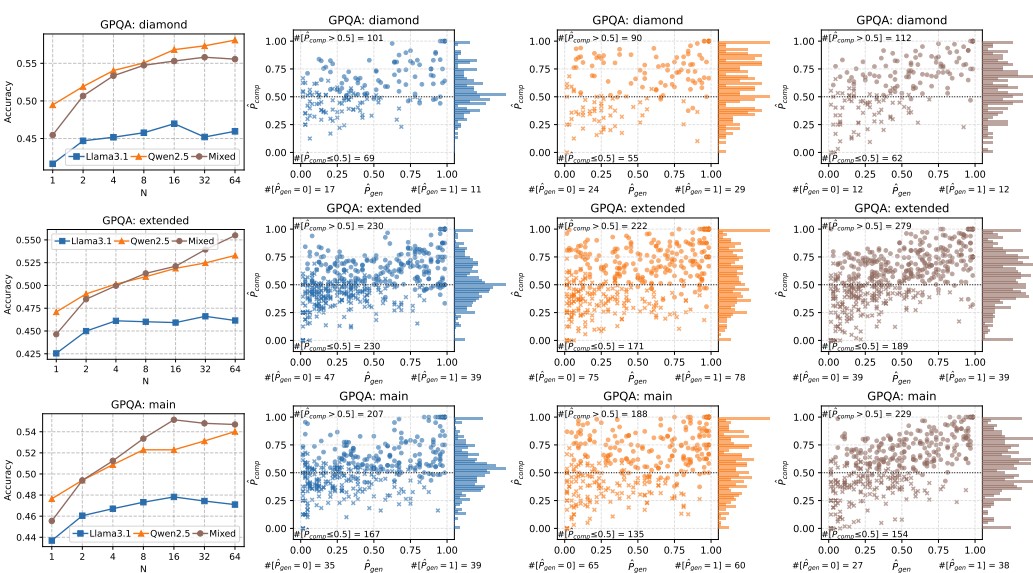

Figure 11: Empirical results of the knockout-style algorithm for each category of GPQA.

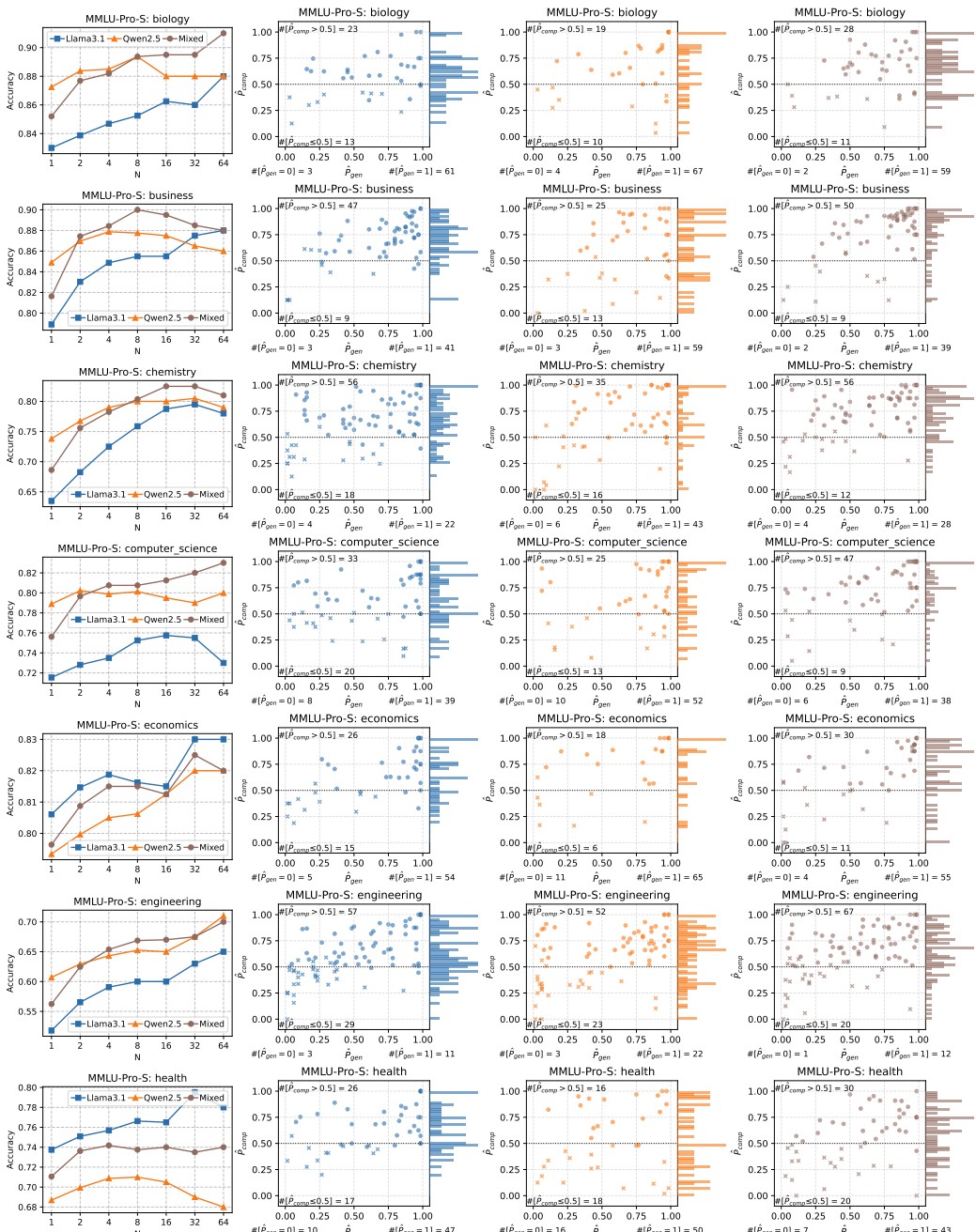

Figure 12: Empirical results of the knockout-style algorithm for each category of MMLU-Pro-S (Part 1).

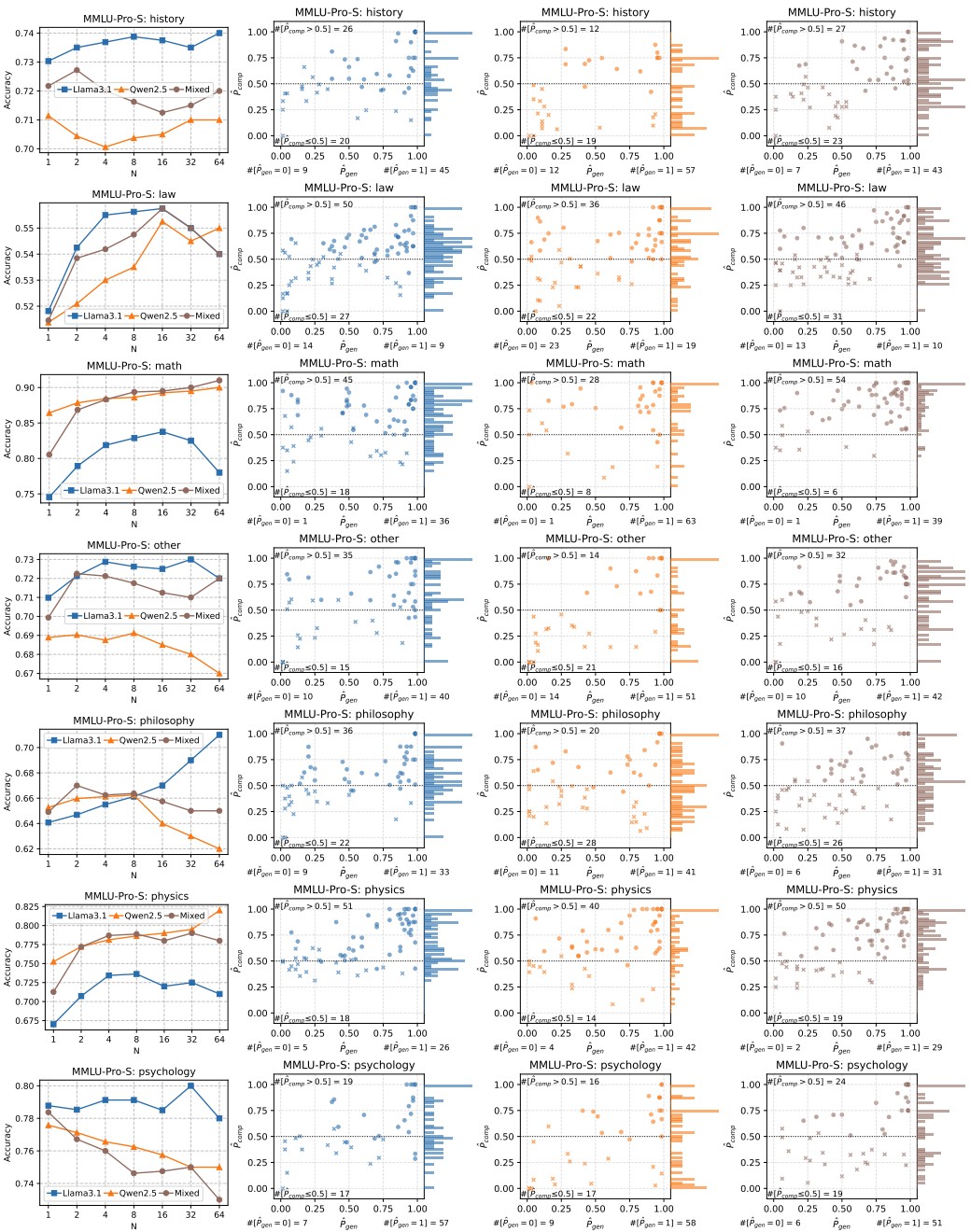

Figure 13: Empirical results of the knockout-style algorithm for each category of MMLU-Pro-S (Part 2).

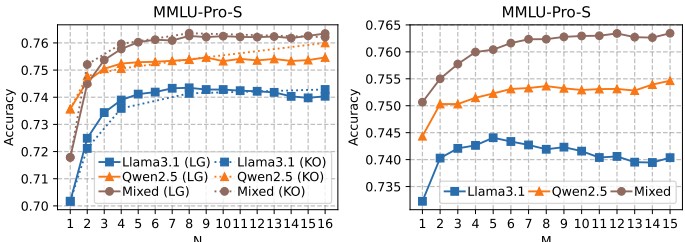

Figure 14: Empirical results for the league-style algorithm on MMLU-Pro-S. **Left:** accuracy versus the number of initial candidates $N$ for the league-style (LG, solid lines) and knockout-style (KO, dotted lines) algorithms, given the same initial candidates. **Right:** accuracy versus $M$, the number of subsampled opponents for each candidate, for the league-style algorithm with $N = 16$.

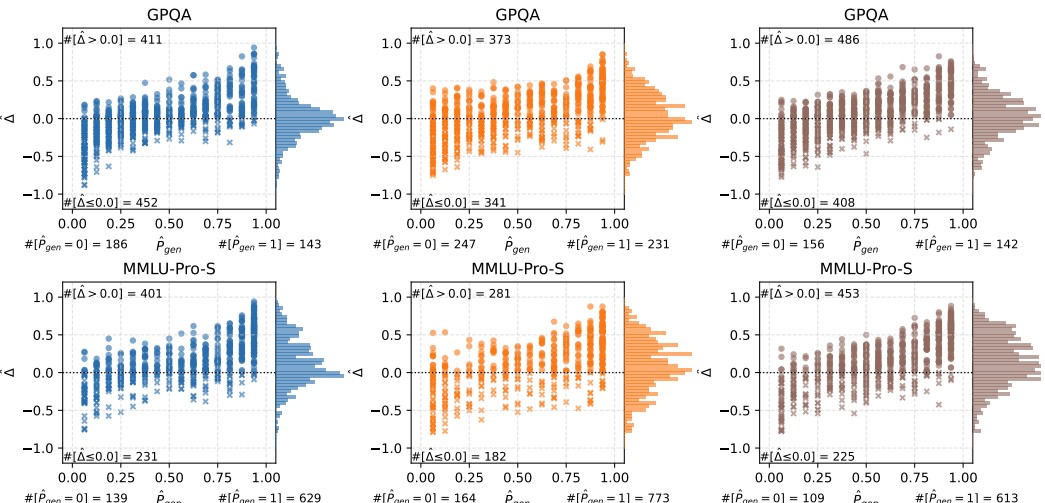

Figure 15: The distribution of GPQA (top) and MMLU-Pro-S (bottom) problems, characterized by $\widehat{p}_{cs}$ and $\widehat{\Delta}$ that are estimated with the empirical results for the league-style algorithm using the Llama3.1 (left), Qwen2.5 (middle) or Mixed (right) option. Each plot is annotated with the number of problems satisfying the condition $\widehat{\Delta} > 0$, $\widehat{\Delta} \le 0$, $\widehat{p}_{cs} = 0$ or $\widehat{p}_{cs} = 1$. To the right of each plot is a histogram for $\widehat{\Delta}$. Each problem is represented by a circle if it is solved correctly by the league-style algorithm with $N = 16$, and by a cross otherwise. We neglect problems with $\widehat{p}_{cs} = 0$ or 1, i.e., problems for which the initial candidate solutions are all incorrect or all correct, since there is no way of obtaining meaningful estimate of $\widehat{\Delta}$ for such problems.

### D.3 Additional results for the league-style algorithm

**Results for MMLU-Pro-S.** Figure 14 includes empirical results for the league-style algorithm on MMLU-Pro-S.

**Distribution of problems.** Figure 15 illustrates the distribution of GPQA and MMLU-Pro-S problems, characterized by $\widehat{p}_{cs}$ and $\widehat{\Delta}$ that are estimated with the empirical results for the league-style algorithm using the Llama3.1 (left), Qwen2.5 (middle) or Mixed (right) option.

**A closer look at both algorithms and their differences.** Figure 16 provides a detailed comparison between the empirical performance of both algorithms. It characterizes the distribution of GPQA and MMLU-Pro-S problems in terms of $\widehat{p}_{comp}$ from the knockout-style algorithm and $\widehat{\Delta}$ from the league-style algorithm, and provides the concrete number of problems that one algorithm solves correctly/incorrectly and the other algorithm solves correctly/incorrectly.

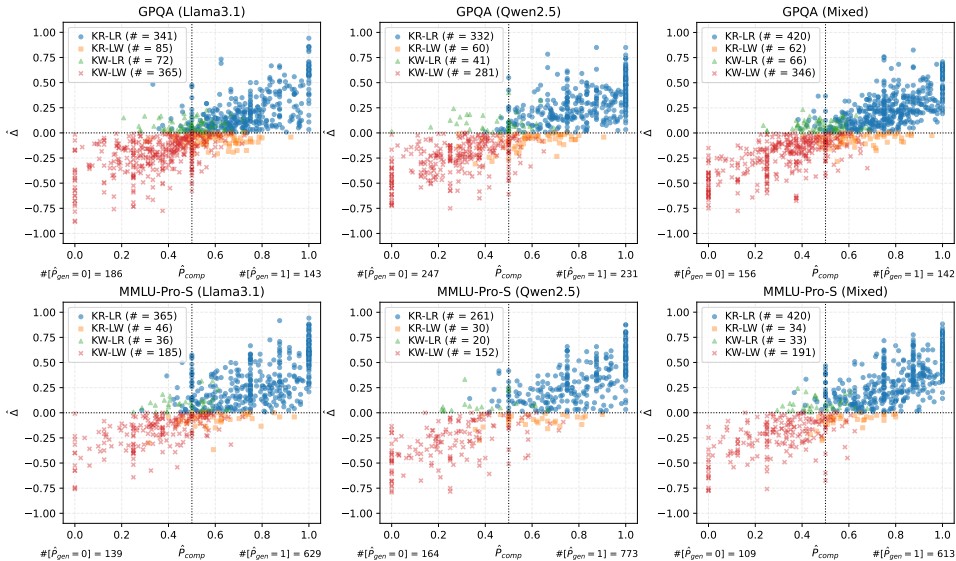

Figure 16: The distribution of GPQA (top) and MMLU-Pro-S (bottom) problems, characterized by $\widehat{p}_{\mathrm{comp}}$ from the knockout-style algorithm and $\widehat{\Delta}$ from the league-style algorithm (both with $N = 16$) using the `Llama3.1` (left), `Qwen2.5` (middle) or `Mixed` (right) option. The following abbreviations are used for the legend: K — knockout, L — league, R — right, W — wrong. For example, "KW-LR (# = 66)" means that there are 66 problems for which the knockout-style algorithm did wrong while the league-style algorithm did right.

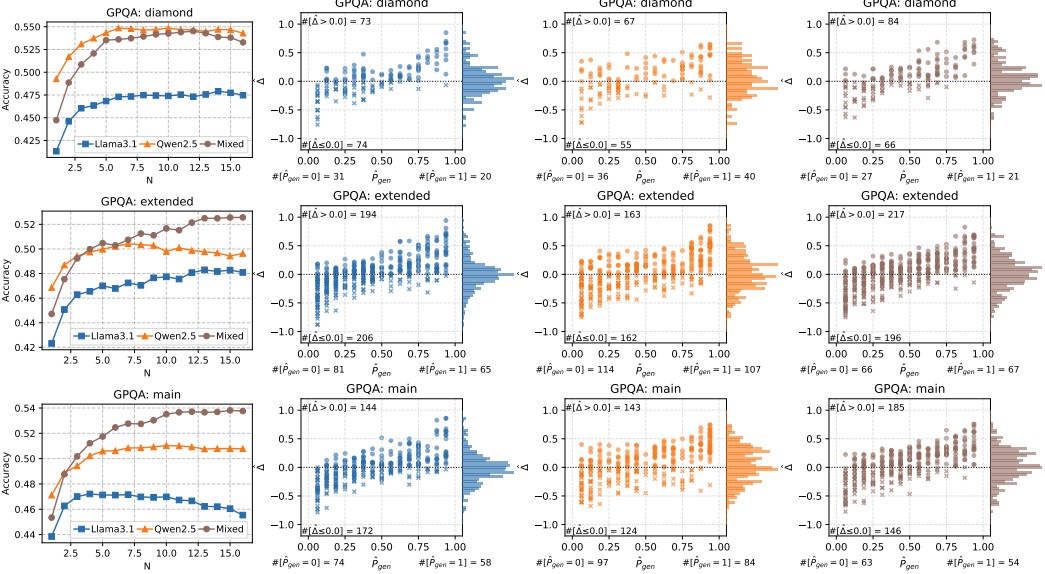

Figure 17: Empirical results of the league-style algorithm for each category of GPQA.

**Results for each category of GPQA and MMLU-Pro-S.** Figure 17 includes empirical results of the league-style algorithm for each category of GPQA, while Figures 18 and 19 include those for MMLU-Pro-S.

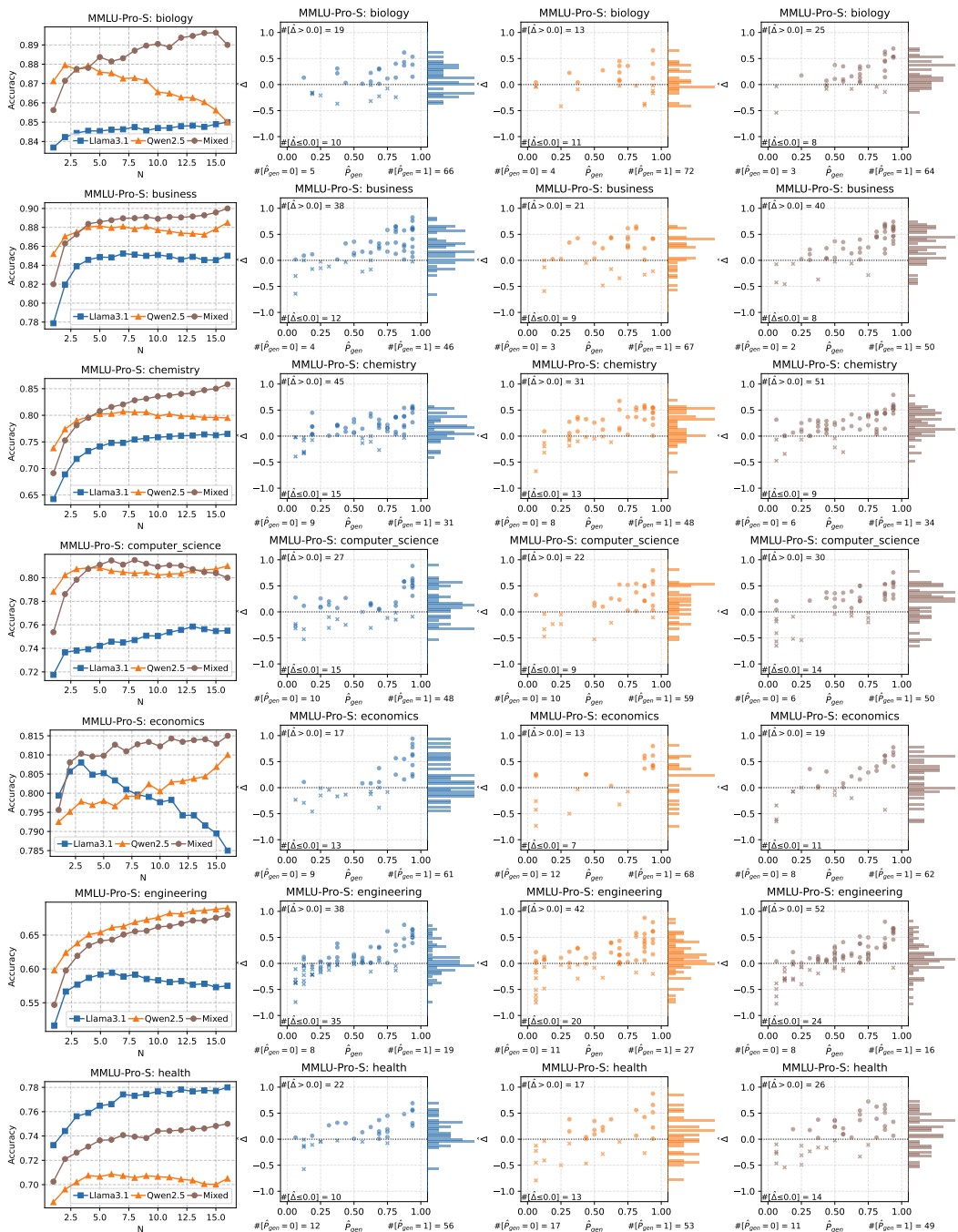

Figure 18: Empirical results of the league-style algorithm for each category of MMLU-Pro-S (Part 1).

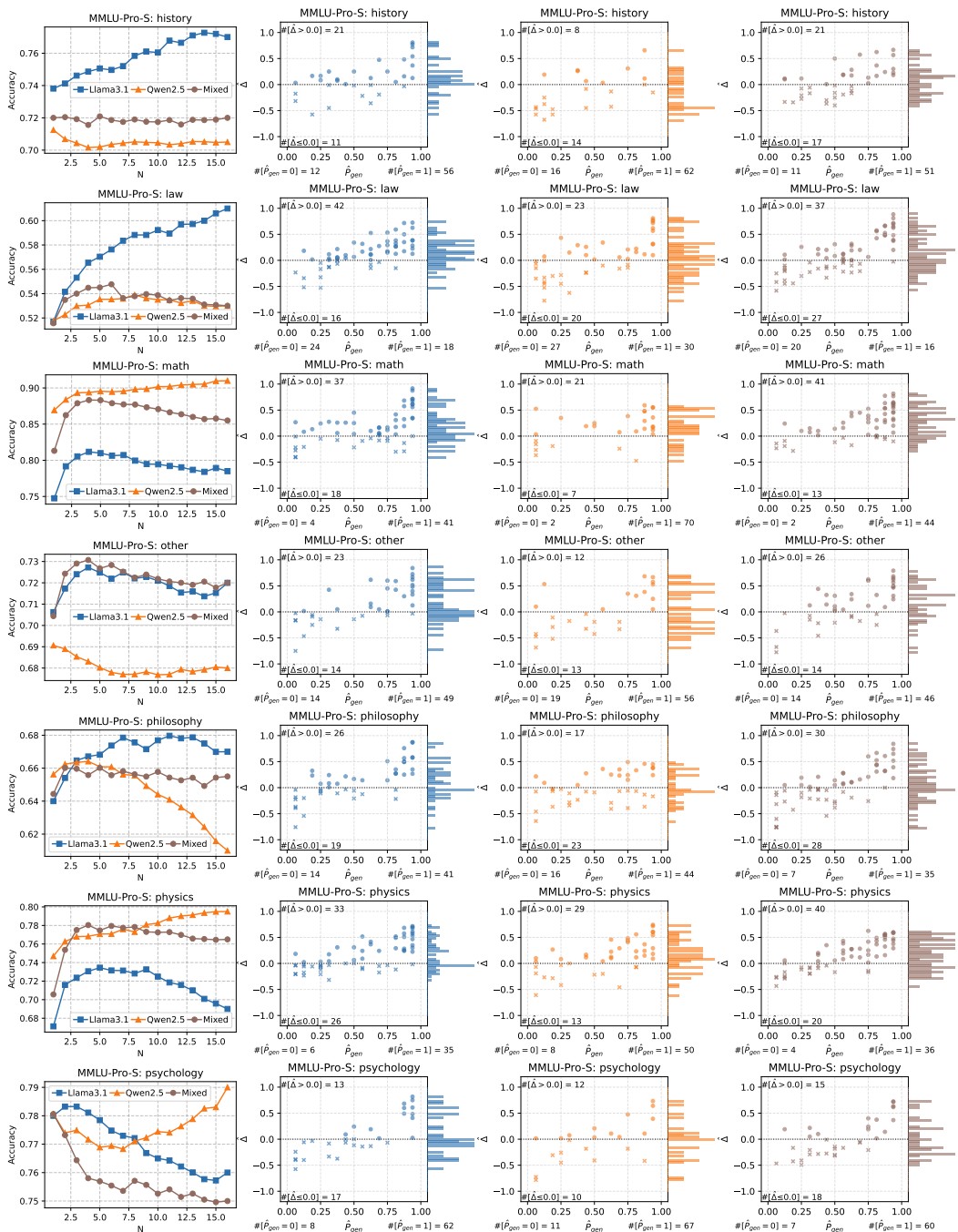

Figure 19: Empirical results of the league-style algorithm for each category of MMLU-Pro-S (Part 2).

Table 1: The adopted prompt template for generating a candidate solution.

% System prompt
Please read the following multiple−choice questions and provide the most likely correct answer based on the options given.

% User prompt
# Question

{question}

# Output Format
```
<reason>your step−by−step reasoning proecss</reason>
<answer>"the answer is (X)" where X is the correct letter choice</answer>
```

Table 2: The adopted prompt template for pairwise comparison.

% System prompt
You are an impartial Judge. Given a question and two candidate solutions, your task is to choose which solution answer the question better. Your judgment should be unbiased, without favoring either Solution 1 or 2.

% User prompt
−−−− QUESTION −−−−
{question}

−−−− Solution 1 −−−−
{candidate_a}

−−−− Solution 2 −−−−
{candidate_b}

−−−− OUTPUT FORMAT −−−−
```
<compare>compare both candidate solutions step−by−step thoroughly, and double check if there are mistakes in either solution</compare>
<winner>Solution 1 or Solution 2 or Tie</winner>
```

