# OpenReview forum: "Provable Scaling Laws for the Test-Time Compute of Large Language Models"
_NeurIPS.cc/2025/Conference — NeurIPS 2025 poster_

### Official Review · Reviewer_NB6t · 2025-06-07

**Clarity:** 2
**Significance:** 3
**Originality:** 3
**Rating:** 4
**Confidence:** 2

**Summary:**

This paper considers scaling laws for test-time compute. In particular, the paper proposes two algorithms to bound the probability that LLMs generate a wrong answer to problems under certain assumptions. The authors provide experiments on multiple choice and math questions to test their algorithms.

**Questions:**

- A algorithm environment describing the algorithm in Section 3 would be helpful.
- What would be some more natural examples when Assumption 3.1 hold, beyond what is discussed in Appendix C? In particular, correct and strong vs correct and weak.
- What's the intuition that we need a separate correct-but-weak set? It's not used in the proof of the Theorem 3.2 in the appendix.
- Assumption 3.1 by itself does not seem to be sufficient to guarantee that the algorithm will fail to choose a correct answer with (exponentially small) vanishing probability. Inherently, there should be an assumption quantifying the win rate between a correct and strong vs a correct and weak answer. It seems be possible such that with constant probability, a correct but weak answer is chosen. Consider the following realistic analogy where the goal is to consider the correctness of proofs (answers) to theorems (questions). Letting an undergrad student be the "aggregator" (who can identify incorrectness, but cannot identify a weak vs a strong answer and thus gives ties to these matches) will yield constant probability of weak answers. The scenario considered in the paper seems to implicitly assume a stronger aggregator who can identify between the three sets (and thus is not black-box).
- Can experiments be provided for larger values of N, e.g. for Figure 3? The current results don't indicate the exponential decay of failure as backed by the experiments.

The fourth point is my major concern in the paper and I may be wrong, but clarifying it will raise my score in the positive light.

**Ethical Concerns:**

["NO or VERY MINOR ethics concerns only"]

**Final Justification:**

The authors provided a thorough response and addressed all of my concerns on Assumption 3.1, so I increased my score. I recommend acceptance, and hope the authors will add this additional clarification on the assumptions to the paper.

**Limitations:**

yes

**Quality:**

3

**Strengths And Weaknesses:**

Strengths:
- The results are backed by complete proofs and experiments.
- The paper is well-written and discussion is well-motivated.

Weaknesses:
- I think the title is a little misleading because the connection with "scaling laws" is rather loose. In particular, the main contribution is introducing algorithms to boost the success rates of LLMs.
- The algorithms and their corresponding theoretical results aren't surprising. For instance, ideas in Section 2 are reminiscient of boosting the success rate of BPP algorithms. Nonetheless, the originality is imposing seemingly reasonable assumptions that apply to LLMs.
- Assumption 3.1 is not motivated very well. Also see the questions below.

---

> ### Author Rebuttal · Authors · 2025-07-31
>
> Thank you for your thoughtful and constructive review!
> We address your questions and concerns in the following.
>
>
> ---
>
>
> # Questions about Assumption 3.1
>
> Thank you for raising the questions about Assumption 3.1, which indicates room for improvement in the clarity of our writing.
> Let us try to explain this assumption better.
>
> **Definitions of "correct-and-strong" and "correct-but-weak" solutions.**
>
> In Assumption 3.1, a "correct-and-strong solution" $y$ is *defined as* a solution that not only contains a correct final answer, but also has a higher average win rate $\mu_{y}$ against a distribution ($M_{gen}$) of solutions than any incorrect solution does.
> On the other hand, a correct-but-weak solution contains a correct final answer, but we make no assumption about its average win rate against $M_{gen}$.
>
> **Is Assumption 3.1 sufficient for success guarantees of the league-style algorithm?**
>
> The answer is yes.
> The main intuition behind the proof of Theorem 3.2 is not about ensuring that the league algorithm will select a correct-and-strong solution as the final output (and this is indeed not guaranteed),
> but rather about ensuring that any incorrect solution will not be selected.
> And this is true because, by definition, the average win rate (against $M_{gen}$) of any incorrect solution is lower than that of a correct-and-strong solution;
> with sufficiently many samples, this is also true for the empirical estimates of average win rates.
>
> Remark 1: We can't say for sure whether the final output of the league algorithm is a correct-and-strong or correct-but-weak solution, since we make no assumption about the average win rate of the latter.
>
> Remark 2: In your review, you mention a potential failure case when a correct-but-weak solution wins with constant probability when competing against a correct-but-strong solution.
> This is not necessarily an issue, since the league algorithm evaluates each candidate solution based on its overall comparison results against multiple opponents, rather than any single one.
>
>
> **Why do we need a separate correct-but-weak set?**
>
> This is mostly because we want to relax as much as possible the technical assumptions required by provable success of the league algorithm.
> Instead of assuming "any correct solution has a higher average win rate (against $M_{gen}$) than any incorrect solution does",
> our final version of Assumption 3.1 says "there exist correct solutions that have higher average win rates than any incorrect solution does", while no assumption is imposed on other correct solutions.
>
> **Natural examples for correct-and-strong vs correct-but-weak solutions?**
>
> Consider a reasoning-focused (e.g., math or physics) multiple-choice question.
> A correct-and-strong solution contains not only a correct final answer (option A/B/C/D) but also a strong reasoning process.
> In contrast, a correct-but-weak solution might contain a flawed reasoning process, but happens to arrive at the correct final answer by luck.
> It is indeed funny that sometimes the LLM could conclude a flawed reasoning process with something like "the result of my calculation is -5, and the options are (A) 100, (B) 200, (C) 300, (D) 400, so I would choose (A) because it is the closest to my calculation".
> We shouldn't expect such a correct-but-weak solution to have a high win rate against another (correct or incorrect) opponent, since the reasoning process in each candidate is taken into account when pairwise comparisons are conducted.
>
> ---
>
> Thank you again for raising the questions.
> We hope that the above clarifications, which will be incorporated into the revised manuscript, have resolved your concerns regarding Assumption 3.1.
>
> ---
>
> # Other questions and comments
>
> **Question about the title, and connection with "scaling laws".**
>
>
> In this work, we adopt the general concept of "inference scaling law", that is,
> "the performance of an LLM or LLM-based system keeps improving with more test-time compute".
> Different metrics of "performance" have been adopted in the literature,
> such as accuracy on a dataset, or success rate on a specific problem;
> the former is indeed equivalent to taking the average of success rates over the problems within a dataset.
> We hope that this can explain why the current work is closely in line with the topic of "inference scaling laws".
>
>
> **Suggestion: "An algorithm environment describing the algorithm in Section 3 would be helpful."**
>
> Thank you for the suggestion. We will add one in our revision.
>
>
>
> **Question: "Can experiments be provided for larger values of N, e.g. for Figure 3?"**
>
>
> In the manuscript, we have chosen the maximum values of N based on the computational resources available to us.
> Following your suggestion, we have extended the results in Figure 3 up to N=256 during the rebuttal period.
> Before we present the updated results, we have to first admit that we mistakenly used the wrong plot for Figure 3 in the manuscript, and we promise to correct it in the revision.
>
> The updated results on GPQA-subset for the "Mixed" option are shown in the following table.
> Recall that in the experiment for Figure 3, we first identify a subset of GPQA problems that the knockout algorithm achieves an estimated $\\hat{p}\_{comp} > 0.5$ on,
> and then plot the accuracy curve for a new, independent trial of the algorithm on this subset.
> The updated results in the first row of the table below show that accuracy of the knockout (KO) algorithm keeps improving with $N$ up to 256, although the trend eventually slows down.
> This is not surprising, because $\hat{p}\_{comp} > 0.5$ is just a proxy for the condition $p\_{comp} > 0.5$, and thus there still remain problems in GPQA-subset that violate Assumption 2.1.
> This motivates us to further tighten the filtering condition, such as $\hat{p}\_{comp} > 0.6$ or $0.7$.
> The results in this table confirm that the accuracy ceiling indeed gets higher on a problem subset that meets a stricter condition.
>
>
> | N | 1 | 2 | 4 | 8 | 16 | 32 | 64 | 128 | 256 |
> | --- | --- | --- | --- | --- | --- | --- | --- | --- | --- |
> | $\hat{p}_{comp} > 0.5$ subset, knockout | 0.556 | 0.657 | 0.700 | 0.728 | 0.752 | 0.766 | 0.782 | 0.786 | 0.797 |
> | $\hat{p}_{comp} > 0.5$ subset, majority | 0.556 | 0.625 | 0.647 | 0.666 | 0.678 | 0.685 | 0.685 | 0.685 | 0.695 |
> | $\hat{p}_{comp} > 0.6$ subset, knockout | 0.609 | 0.728 | 0.775 | 0.807 | 0.834 | 0.849 | 0.863 | 0.865 | 0.881 |
> | $\hat{p}_{comp} > 0.6$ subset, majority | 0.609 | 0.698 | 0.718 | 0.735 | 0.745 | 0.752 | 0.749 | 0.747 | 0.755 |
> | $\hat{p}_{comp} > 0.7$ subset, knockout | 0.657 | 0.788 | 0.837 | 0.871 | 0.893 | 0.907 | 0.917 | 0.926 | 0.931 |
> | $\hat{p}_{comp} > 0.7$ subset, majority | 0.657 | 0.753 | 0.774 | 0.788 | 0.798 | 0.803 | 0.804 | 0.803 | 0.809 |
> | $\hat{p}_{comp} > 0.8$ subset, knockout | 0.730 | 0.869 | 0.909 | 0.932 | 0.945 | 0.960 | 0.964 | 0.970 | 0.976 |
> | $\hat{p}_{comp} > 0.8$ subset, majority | 0.730 | 0.841 | 0.862 | 0.877 | 0.894 | 0.900 | 0.905 | 0.899 | 0.899 |
>
>
>
>
>
> ---
>
> Thank you again for your review, which helps us a lot in improving the current work!
> We hope that our responses have provided satisfactory answers to your questions,
> but do let us know if you have any further concern that we can address.

---

> > ### Comment · Reviewer_NB6t · 2025-08-02
> >
> > I thank the authors for clarifying my concern on Assumption 3.1.

---

> > > ### Author Response · Authors · 2025-08-04
> > >
> > > Thank you for your reply!
> > > We are glad to know that your major concern on Assumption 3.1 has been clarified.
> > >
> > > We look forward to your updated and finalized evaluation of our work.
> > > If you need more information before making your final decision,
> > > we are always open to discussions :)

---

> > > > ### Comment · Reviewer_NB6t · 2025-08-05
> > > >
> > > > After reading the discussion on the assumptions from the other reviews and replies, I have another question regarding the verifiers $\mathcal{M}_{comp}$. Can we say that the verifier required for Assumption 3.1 doesn't need to be as powerful as one for Assumption 2.1?
> > > >
> > > > It would be useful to discuss additional instances when Assumption 3.1 holds, especially when answers do not take support in a discrete set. It appears all the experiments all are in the setting where answers are discrete-valued. That being said, I'm not familiar with guarantees for test-time compute, so I'm not sure whether this is a shared limitation from previous works.

---

> > > > > ### Author Response · Authors · 2025-08-06
> > > > >
> > > > > Thank you for your reply! We answer your questions in the following.
> > > > >
> > > > > ---
> > > > >
> > > > > **Can we say that the (pairwise) verifier $M_{comp}$ required for Assumption 3.1 doesn't need to be as powerful as one for Assumption 2.1?**
> > > > >
> > > > >
> > > > > The short answer is "yes and no", and let us explain.
> > > > >
> > > > >
> > > > > Assumption 2.1 requires that the pairwise verifier $M_{comp}$ is able to distinguish (with $p_{comp} > 0.5$) between *any pair* of candidate solutions where one of them is correct and the other is incorrect.
> > > > > In an attempt to mitigate such a requirement on $M_{comp}$,
> > > > > Assumption 3.1 only requires that with $M_{comp}$,
> > > > > there exist correct candidates whose *average win rates* (against a distribution of opponents, namely $M_{gen}$) are higher than that of any incorrect candidate.
> > > > > In this sense, we may say that $M_{comp}$ required for Assumption 3.1 doesn't need to be as powerful as one for Assumption 2.1.
> > > > >
> > > > >
> > > > >
> > > > > However, to rigorously prove that Assumption 3.1 is weaker than 2.1, we need further conditions. One possibility would be the following:
> > > > >
> > > > > **Assumption X (informal):** The pairwise verifier $M_{comp}$ has no strong preference when comparing a pair of candidates $y_1, y_2$ that are both correct or both incorrect,
> > > > > i.e., $P_{r \sim M_{comp}(x, y_1, y_2)}(r \text{ identifies } y_1 \text{ as the winner}) \approx 0.5$.
> > > > >
> > > > > **Claim:** Assumption 2.1 (with parameter $p_{comp}$) and Assumption X together imply Assumption 3.1 with parameter $\Delta \approx p_{comp} - 0.5 > 0$.
> > > > >
> > > > > **Proof:** Let us denote by A (resp. B) the set of all possible correct (resp. incorrect) solutions.
> > > > > Define $p_A = P_{y \sim M_{gen}(x)}(y \in A)$, and similarly for $p_B$.
> > > > > Then we can calculate the average win rate of a particular candidate:
> > > > > for $y \in A$, we have $\mu_{y} \approx p_A \times 0.5 + p_B \times p_{comp}$,
> > > > > whereas for $z \in B$, we have $\mu_{z} \approx p_A \times (1 - p_{comp}) + p_B \times 0.5$.
> > > > > We thus have $\mu_{y} - \mu_{z} \approx p_{comp} - 0.5 > 0$, which implies Assumption 3.1 and concludes this proof.
> > > > >
> > > > > Of course, Assumption X is just one possibility, and there can be other options.
> > > > > On the other hand, without any assumption regarding the verifier's preference in comparing a pair of candidates that are both correct or both incorrect,
> > > > > Assumption 2.1 alone does not necessarily imply Assumption 3.1.
> > > > > Example C.3 in the appendix is one such counterexample, where an incorrect candidate B somehow attains the highest average win rate, due to its extremely high win rate (0.9) against another incorrect candidate C.
> > > > >
> > > > >
> > > > >
> > > > > We hope that the above discussions have offered useful intuitions for Remark 3.3 and the examples in Appendix C that explain why neither Assumptions 2.1 or 3.1 is strictly weaker than the other.
> > > > >
> > > > >
> > > > > ---
> > > > >
> > > > > **It would be useful to discuss additional instances when Assumption 3.1 holds, especially when answers do not take support in a discrete set. It appears all the experiments are in the setting where answers are discrete-valued.**
> > > > >
> > > > >
> > > > > We would like to highlight that all assumptions, algorithms and theoretical guarantees proposed in this work are **not** limited to settings where answers are discrete-valued.
> > > > > While some prior works on guarantees of test-time scaling, e.g., for majority voting, require discrete-valued answers (by the nature of the algorithm, as discussed in Lines 340 - 342),
> > > > > our current work is free from such a limitation.
> > > > >
> > > > >
> > > > > For an additional instance when Assumption 3.1 holds, the "Proof" part in our response to the previous question indeed serves as an example, which might be regarded as a generalization of Example C.1.
> > > > > We plan to add it to Appendix C in our revision.
> > > > > In particular, note that A (resp. B) in the above example is defined as the set of all possible correct (resp. incorrect) candidate solutions, whose final answers do not have to take support in a discrete set.
> > > > > For example, a final answer would be a piece of code in a competitive coding task,
> > > > > which is considered correct if it passes all test cases;
> > > > > or, it would be a summary in a summarization task,
> > > > > which is considered correct if it is coherent and concise while containing all the key points of the original text.
> > > > >
> > > > >
> > > > > In the current work, we have chosen to stick with multiple-choice or math questions in our experiments, mostly for the purpose of (1) unambiguous and objective evaluation, and (2) controllable computational costs of experiments.
> > > > > We hope that our work will inspire the community to try out the proposed algorithms in more diverse settings (especially those where answers are not discrete-valued), although this is beyond the scope of our current focus in this work.
> > > > >
> > > > >
> > > > > ---
> > > > >
> > > > > Thank you again for the thoughtful questions, which have helped us improve this work!
> > > > > We hope that our responses can assist you in finalizing your evaluation,
> > > > > and we are still open to discussions if you have any further question.

---

> > > > > > ### Author Response · Authors · 2025-08-08
> > > > > > **Follow-up**
> > > > > >
> > > > > > Dear Reviewer NB6t,
> > > > > >
> > > > > > As the reviewer-author discussion period is ending soon, we would like to know if our previous response has provided satisfactory answers to your additional questions regarding Assumption 3.1.
> > > > > >
> > > > > > Thank you!
> > > > > >
> > > > > > Best,
> > > > > >
> > > > > > Authors

---

> > > > > > > ### Comment · Reviewer_NB6t · 2025-08-08
> > > > > > >
> > > > > > > Thank you for addressing my questions. I am more convinced of the novelty of the assumptions now and will increase my score accordingly.

---

> > > > > > > > ### Author Response · Authors · 2025-08-09
> > > > > > > >
> > > > > > > > Thank you for your reply and updated evaluation! We are glad to know that your questions have been resolved.

---

### Official Review · Reviewer_SKdM · 2025-06-25

**Clarity:** 3
**Significance:** 3
**Originality:** 4
**Rating:** 4
**Confidence:** 2

**Summary:**

The authors propose two theoretically grounded test-time inference scaling algorithms: knockout-style and league-style, which are designed to improve the reasoning accuracy of LLMs under increased computational budgets. Theoretical analyses demonstrate that, under the assumptions that the LLM has a non-zero probability of generating a correct solution and can make pairwise comparisons better than random guessing, the failure probability of both algorithms decreases exponentially or polynomially as test-time compute increases.

**Questions:**

Please see weakness.

**Ethical Concerns:**

["NO or VERY MINOR ethics concerns only"]

**Limitations:**

yes

**Paper Formatting Concerns:**

no concerns

**Quality:**

3

**Strengths And Weaknesses:**

Strengths

1. The paper presents two inference scaling algorithms with rigorous theoretical guarantees, demonstrating solid theoretical foundations.
2. The proposed methods rely solely on a black-box LLM, requiring no external verifier or reward model, which simplifies implementation.
3. The paper is clearly written and well-structured.

Weaknesses

1. The citation at line 30 appears abrupt and lacks sufficient explanation or context.
2. It is unclear how the authors determine the values of N and K. What are the implications for time efficiency in real-world applications?
3. The paper does not seem to compare against strong baseline methods. Are there no suitable test-time inference scaling approaches available for comparison?
4. The datasets used in the experiments are mostly multiple-choice question types. Are there evaluations on other types of tasks?
5. The paper lacks an in-depth failure case analysis. For example, why do some problems remain unsolved even when more candidates are generated?
6. What is the difference between the proposed knockout-style algorithm and the BoN sampling method?

---

> ### Author Rebuttal · Authors · 2025-07-31
>
> Thank you for your thoughtful and constructive review!
> We address your questions and concerns in the following.
>
>
> ---
>
>
>
> **Question 1: The citation at line 30 appears abrupt and lacks sufficient explanation or context.**
>
> Thank you for pointing this out.
> We will divide the cited papers into smaller sub-categories and provide more context about this line of literature in our revision.
>
> ---
>
> **Question 2: It is unclear how the authors determine the values of N and K. What are the implications for time efficiency in real-world applications?**
>
>
> **In terms of experiments:**
> Detailed explanations for the choices of N and K in our experiments can be found in Lines 223 - 226,
> which we reiterate in the following.
> For the knockout algorithm, we fix K at a small value (2 or 4), and scale up N alone;
> the total number of LLM calls thus grows linearly with N.
> For the league algorithm, we scale up N and mostly adopt a round-robin setting where each candidate is compared against all other candidates, leading to a total of $O(N^2)$ LLM calls;
> nonetheless, our experiment results in Lines 329 - 336 (as well as Theorem 3.2) also confirm that it is feasible to use a significantly smaller number of opponents (thus fewer LLM calls) while maintaining accuracy.
>
>
> **In terms of methodology and theory:**
> For the knockout algorithm, our general recommendation is scaling up N alone while fixing K at a small value (mainly to account for the positional bias of LLMs and to cover multiple LLMs if a mixture of them is used in the algorithm), which is backed by Theorem 2.3.
> For the league algorithm, our general recommendation is scaling up N alone and adopting the round-robin setting for simplicity, although it is certainly feasible to tune down the number of opponents for better efficiency.
> In practice, one may scale up N until a given budget is met, or use the "anytime" variants of the proposed algorithms that can automatically adapt to the compute budget unknown a priori (cf. Lines 627 - 640).
>
> It is also worth noting that the LLM calls in either algorithm are highly parallelizable (as explained in Lines 147 - 156 for knockout, and Lines 170 - 171 for league), hence the wall-clock latency can substantially benefit from parallel computation.
>
> ---
>
> **Question 3: The paper does not seem to compare against strong baseline methods. Are there no suitable test-time inference scaling approaches available for comparison?**
>
> Given the vast literature on test-time scaling techniques,
> we have chosen to focus our attention on methods that are
> (1) within the category of "repeated sampling and then aggregation" (as explained in Lines 28 - 31), and
> (2) foundational and broadly applicable to generic tasks, rather than task-specific.
> To the best of our knowledge, majority voting (or "self-consistency") and Best-of-N are the most prominent approaches satisfying these conditions, and they are most widely adopted in benchmarking LLMs.
> Furthermore, as discussed in Lines 349 - 351, we refrain from comparing our methods with BoN in our experiments, since introducing an external verifier or reward model will bring extra variability that makes it difficult to conduct a fair and meaningful empirical comparison.
> This leaves us with majority voting as the only suitable baseline for our empirical study.
>
> ---
>
>
> **Question 4: The datasets used in the experiments are mostly multiple-choice question types. Are there evaluations on other types of tasks?**
>
> We choose to stick with multiple-choice or math questions in the current work, for the purpose of
> (1) unambiguous and objective evaluation, and (2) controllable computational costs of experiments.
> We hope that our work will inspire the community to try out the proposed algorithms in more diverse tasks, although this is beyond the scope of our current focus in this work.
>
>
> ---
>
> **Question 5: The paper lacks an in-depth failure case analysis. For example, why do some problems remain unsolved even when more candidates are generated?**
>
> Indeed, a large part of our empirical study has been dedicated to understanding the empirical success and failure of the proposed algorithms, and in particular, to what extent the proposed technical assumptions hold true in reality.
> This includes Lines 251 - 308 for the knockout algorithm, Lines 316 - 328 for the league algorithm, and the corresponding figures of experiment results.
> For example, if a problem (e.g., "solve the P versus NP problem") is simply beyond the LLM's capabilities, then the LLM will effectively have $p_{gen} = 0$ or $p_{comp} \le 0.5$ on this problem, which violates Assumption 2.1 required by Theorems 2.2 and 2.3.
> In such cases, it is no surprise that the knockout algorithm (or other test-time scaling methods) fails to solve this problem.
>
>
> ---
>
>
> **Question 6: What is the difference between the proposed knockout-style algorithm and the BoN sampling method?**
>
> Both methods initially generate multiple candidate solutions.
> The difference lies in the aggregation stage that returns a final output.
> BoN uses an external verifier or reward model to score each candidate independently, and then pick the one with the highest score.
> In contrast, the knockout-style algorithm conducts a series of pairwise comparisons among the candidates; this is largely motivated by the common belief (which is backed by extensive empirical evidence) that it is typically much easier to compare a pair of solutions than to assign an accurate absolute score for each individual solution.
>
>
>
>
> ---
>
> Thank you again for your review, which helps us a lot in improving the current work!
> We hope that our responses have provided satisfactory answers to your questions,
> but do let us know if you have any further concern that we can address.

---

> > ### Comment · Reviewer_SKdM · 2025-08-04
> >
> > The authors have addressed my concerns, and I maintain my positive score.

---

### Official Review · Reviewer_fB8B · 2025-06-30

**Clarity:** 4
**Significance:** 3
**Originality:** 3
**Rating:** 5
**Confidence:** 4

**Summary:**

The authors propose two knockout style algorithms to improve the likelihood of generating a correct answer from a large language model.
In the first algorithm they generate N possible answers and then select the winner through a bracket tournament system where a judge model picks a winner by performing repeated pairwise comparison.
In the second algorithm, each generated answer gets compared against K other randomly selected answers and the answer with the highest average winrate gets picked.
The authors then prove probabilistic success guarantees for both algorithms as N & K are scaled under simple theoretical assumptions.
Finally, they show that their algorithm improves the success rate of LLMs across three multiple choice datasets.

**Questions:**

- As noted in the weaknesses above I would have liked to have seen a further acknowledgement that the assumptions are quite strong in practice earlier in the paper. I think this would take the paper from a weak accept to an accept for me.
- Did you investigate the possiblity of the existance of "strong-and-incorrect" candidates in your empirical evaluation? If so could you add some discussion about this in the paper?
- I'm quite interested in the results of MMLU-Pro-S philosophy shown in figure 4, where it seems that increasing N can be actively harmful to accuracy for certain models, how robust is this result? Is this simply because the dataset is quite small and thus quite noisy or is this a reliably reproducible result?

**Ethical Concerns:**

["NO or VERY MINOR ethics concerns only"]

**Final Justification:**

Authors successfully addressed all my questions and concerns so raising score to accept.

**Limitations:**

yes

**Quality:**

2

**Strengths And Weaknesses:**

Strengths:
 - The proposed algorithms are very simple/elegant and thus easy to implement
 - The paper clearly outlines a compelling reason for why this method would work
 - The authors provide formal guarantees which aren't often available for LLMs
 - The empirical evaluation section is thorough and left me convinced that this method significantly improves the performance of models on multiple choice tasks.
Weaknesses:
 - The assumptions the authors make for both of their algorithms are much stronger than they might initially appear.
    1. For the bracket algorithm the assumption of p_comp > 0.5 is key to the exponentially decaying rate as we scale N & K, but this assumption is implausible unless the success probabilities from the judge model are independent of each other as the candidate solutions are varied.
    2. For the two-stage league-style algorithm the authors assume the existence of "correct-and-strong" answers whose expected winrate score when compared to the expected winrate of incorrect algorithms is greater than 0. But if a single "incorrect-and-strong" answer exists in the candidate generation set then the proof no longer works.
The paper would be significantly improved if the authors made the limitations of these assumptions much clearer earlier on in the paper.
- I would have liked to have seen error bars for each of the datapoints in the scaling plots to get a better idea of the variance of these results, especially for figures that represent smaller datasets.

---

> ### Author Rebuttal · Authors · 2025-07-31
>
> Thank you for your thoughtful and constructive review!
> We address your questions and concerns in the following.
>
>
> ---
>
>
> **Suggestion: "The paper would be significantly improved if the authors made the limitations of these assumptions much clearer earlier on in the paper."**
>
>
> Thank you for pointing out some limitations of the proposed assumptions, which helps enhance our understanding of them.
> We acknowledge that these assumptions, as sufficient conditions for provable success of the proposed algorithms in theory, can still look strong from a more practical perspective.
> It would be intriguing future work to push forward the boundary of our knowledge about what sufficient and/or necessary conditions make provable inference scaling laws possible.
>
>
> Following your suggestion, we plan to make these revisions to the current work:
>
> - In Line 58, we will replace "two natural (and arguably minimal) assumptions" with simply "two assumptions".
>
> - In Lines 64 - 65 (the second paragraph of "main contributions" in introduction), we will add one sentence to state our assumption for the league algorithm more explicitly: "there exist correct solutions whose average win rates against a distribution of opponents are higher than that of any incorrect solution".
>
> - In Line 71, we will slightly rephrase the last sentence of introduction, to faithfully acknowledge the gap between theory and practice. A tentative version is as follows: "Although the technical assumptions in our theories can seem strong from a more practical perspective, our empirical results confirm that the proposed algorithms, developed based on the theoretical insights, indeed perform well and demonstrate outstanding scaling properties across diverse LLMs and datasets."
>
> - In Lines 112 - 114 (following the statement of Assumption 2.1), we will remark that $p_{comp} > 0.5$ is assumed for any pair of solutions, which is a strong assumption and thus will motivate our development of the league algorithm (and its theory) in the next section. Such a remark indeed resembles the beginning of Section 3 in the current manuscript.
>
> - In Lines 177 - 181 (following the statement of Assumption 3.1), we will remark that Assumption 3.1 precludes the existence of an adversarial "incorrect-but-strong" candidate that achieves a higher average win rate than any correct solution does.
>
> We hope that these will make the strengths and limitations of Assumptions 2.1 and 3.1 clearer to the readers,
> and we'd love to hear your feedback on whether any further improvement can be made.
>
>
> **Question: "Did you investigate the possibility of the existence of "incorrect-and-strong" candidates in your empirical evaluation? If so could you add some discussion about this in the paper?"**
>
> Yes, we have seen such candidates in our experiments.
> Recall that in Lines 318 - 320, we define $\hat{\Delta}$ as the difference between the empirical average win rate of the strongest correct candidate, and that of the strongest incorrect candidate.
> The scatter plot in Figure 5, as well as those in Appendix D.3, indicates the existence of problems with $\hat{\Delta} < 0$;
> in other words, for each of these problems, there exists an incorrect-but-strong candidate that achieves a higher average win rate than any correct candidate does.
>
> We will add a discussion in the paragraph of Lines 322 - 328 about this, to explain that the existence of incorrect-but-strong candidates (which violates Assumption 3.1) prevents the league algorithm from boosting the accuracy up to 100% on a dataset, similar to the case of the knockout algorithm that has been discussed in Lines 251 - 291.
>
>
> **Question: error bars for each datapoint in the scaling plots?**
>
>
> Due to the high computational or monetary costs of the experiments,
> we have run the knockout or league algorithm only once for each (model, dataset) combination,
> which prevents us from plotting error bars in the scaling plots.
> Nonetheless, we have made our best efforts to enhance the stability and reliability of the plots in the manuscript:
>
> - For the knockout algorithm, we take advantage of its binary tree structure (shown in Figure 1).
> For example, after running the algorithm once with N=64, we automatically get the results of 64 independent trials for N=1, 32 trails for N=2, 16 trials for N=4, and so on.
> We have thus taken the average of accuracy values from multiple independent trials for each datapoint (except for the rightmost one) in each scaling curve.
>
> - This is also true for the league algorithm.
> For example, after running the algorithm once with N=16, we are able to obtain the results of multiple trials for N=8 (or any value smaller than 16),
> each corresponding to 8 randomly sampled candidate solutions and the comparison results among them.
> Each datapoint (except for the rightmost one) in each scaling curve has been calculated by an average of multiple results obtained this way.
>
>
> **Question: how robust is the result for MMLU-Pro-S Philosophy shown in Figure 4?**
>
>
> Figure 4 indeed suggests that increasing N can be harmful for Qwen2.5 on a 100-question subset (cf. Line 207) of MMLU-Pro-Philosophy.
> To see whether this is a robust and reproducible result,
> we conduct an experiment on the full MMLU-Pro-Philosophy set, which contains 499 questions, during the rebuttal period.
> The result demonstrates a similar trend, as shown in the following table:
>
> | N                                 | 1 | 2 | 4 | 8 | 16 | 32 | 64 |
> | --- | --- | --- | --- | --- | --- | --- | --- |
> | Accuracy by knockout             | 0.630 | 0.630 | 0.632 | 0.630 | 0.627 | 0.621 | 0.615 |
> | Accuracy by majority      | 0.630 | 0.629 | 0.633 | 0.635 | 0.632 | 0.632 | 0.627 |
>
> As discussed in Lines 306 - 308, our intuition for this phenomenon is that pairwise comparisons among candidate solutions might bring limited benefits for knowledge-heavy questions that mostly require correct memorization of relevant knowledge.
> The corresponding scatter plot in Figure 13 (5th row, 3rd column) indeed shows that there are slightly more problems with $\hat{p}_{comp} < 0.5$ than with $\\hat{p}_{comp} > 0.5$ for Qwen2.5 on the MMLU-Pro-Philosophy subset.
>
>
>
>
>
> ---
>
> Thank you again for your review, which helps us a lot in improving the current work!
> We hope that our responses have provided satisfactory answers to your questions,
> but do let us know if you have any further concern that we can address.

---

> > ### Comment · Reviewer_fB8B · 2025-08-03
> >
> > Thank you for your detailed response. My concerns have been addressed and I will update my score accordingly.

---

> > > ### Author Response · Authors · 2025-08-04
> > >
> > > Thank you for your reply! We are glad to know that your concerns have been addressed.

---

### Official Review · Reviewer_PgPW · 2025-07-03

**Clarity:** 3
**Significance:** 2
**Originality:** 2
**Rating:** 3
**Confidence:** 4

**Summary:**

The paper proposes and studies two algorithms for provable scaling laws for the test-time compute. The first algorithm, the two-stage knockout-style algorithm, first generates multiple candidate solutions and then aggregate them via knockout tournament. The second algorithm, the two-stage league-style algorithm, first generate multiple candidates and then each candidate is evaluated by its average win rate against multiple opponents. The authors prove scaling laws with respect to test-time compute and run empirical experiments to validate the proposed theories.

**Questions:**

I am mainly concerned about the assumptions. I will raise my score if the authors can well-justify the assumptions. Specifically, the authors claim the drawback of BoN is requirement of a perfect verifier while the assumption the authors require for their proposed algorithms also implies a perfect verifier. Although the “perfectness” assumption can be relaxed for two-stage  league-style algorithm, this still is a strong assumption. It just changes what we mean by “perfectness”. Although two-stage league-style algorithm is more robust, adversarial candidate should still be able to break the assumption.

Also, regarding definition 1.1, the authors define the concept of provable inference scaling law to a specific input problem. So, this property of provable inference scaling law is not only a property of the algorithm but also a property of a specific input data. I feel this definition that includes the data is less useful than a definition that does not involve the data.

**Ethical Concerns:**

["NO or VERY MINOR ethics concerns only"]

**Final Justification:**

My primary concern remains with the assumptions presented. While I agree with the authors that these assumptions are weaker compared to those required by BoN, they are still quite strong, and the improvement over BoN is relatively minor. Nonetheless, the authors have effectively clarified my other concern regarding the fairness of their comparison. Additionally, their proposed revisions to the wording surrounding the assumptions have improved the paper's clarity and rigor. Consequently, I have raised my score accordingly.

**Limitations:**

The authors have included a limitation section.

**Quality:**

2

**Strengths And Weaknesses:**

Overall, the idea of provable scaling laws for test-time compute is interesting. The experiments result also demonstrate that the proposed algorithms outperform baseline methods such as majority voting. However, I am concerned about the assumptions. Assumption 2.1 and assumption 3.1 are quite strong.

For example, I see assumption 2.1 technically the same as having a perfect reward model. The reason is that it requires that the large language model can distinguish the right solution to ALL the incorrect solutions. However, as one scales up more candidates, it is likely that there might be an adversarial solution that can fool the LLM (the same failure mode of BoN) which results in $p_{comp}$ smaller than 0.5.

Indeed, the authors find in experiments that there are quite some inputs that do not satisfy this assumption (Fig 3 and Fig 5) and as a result, there is no provable scaling law for those problems. Additionally, for MATH-500 results in Fig. 6, the results suggest that the know-out algorithm does not have a provable inference scaling law for this dataset.


Additionally, I am concerned about the comparison between the proposed algorithms and the baseline of majority voting. Specifically, the authors only count the compute to generate the candidates but neglect the compute for aggregation. This is an unfair comparison as the authors use CoT for aggregation and as a result the compute for aggregation is non-trivial and should be accounted for.

---

> ### Author Rebuttal · Authors · 2025-07-31
>
> Thank you for your thoughtful and constructive review!
> We address your questions and concerns in the following.
>
>
> ---
>
>
> # Concerns about Assumptions 2.1 and 3.1
>
>
> Thank you for pointing out some limitations of the proposed assumptions, which helps enhance our understanding of them.
> One of our objectives in this work is to push forward the boundary of our knowledge about what sufficient and/or necessary conditions make provable inference scaling laws possible.
> This work presents our best progress so far, and with some theoretical insights, we have developed algorithms that indeed perform well empirically.
> Nonetheless, we do acknowledge that the proposed assumptions (as sufficient conditions for theoretical guarantees) can still look strong from a more practical perspective, and it would be intriguing future work to further relax the assumptions for provable inference scaling laws.
>
>
> ## Justification: why the proposed assumptions improve upon what is required by BoN
> The reviewer points out that Assumptions 2.1 and 3.1 can be fragile and broken by a single adversarial candidate solution,
> akin to the case of BoN.
> We agree that this is indeed a limitation of the proposed assumptions, and plan to revise the manuscript (which will soon be elaborated) to better reflect this.
>
> Meanwhile, we also believe that Assumptions 2.1 and 3.1, as sufficient conditions for the provable success of our proposed algorithms, are strictly weaker than what is required by BoN.
> The key lies in the difference between *pointwise verification/scoring* and *pairwise comparison*.
> It is widely believed that the latter is typically much easier and more accurate than the former,
> supported by common practice and extensive empirical evidence.
>
> We claim that this should be true in theory as well;
> in particular, "no adversarial candidate exists for BoN" implies "no adversarial candidate exists for the knockout algorithm", but the reverse is not necessarily true.
> To see this, note that a naive bottomline for pairwise comparison would be simply to verify/score each candidate within the pair separately (in a pointwise manner), and then compare the results to select the winner.
> In this case, it is easy to check that if BoN enjoys success guarantees, then so does the knockout algorithm.
> As for the reverse direction of the above claim, note that pairwise comparison allows putting two candidate solutions side by side within the prompt, which is *strictly more informative* than looking at either candidate alone (as done in pointwise verification for BoN).
>
> Another benefit of our proposed assumptions is their probabilistic nature.
> This allows the proposed algorithms to amplify the success probability up to 1, while tolerating (a possibly massive number of) errors in comparing pairs of candidate solutions, as can be seen from the proofs of Theorems 2.3 and 3.2.
>
>
> Given the above, we believe that the proposed Assumptions 2.1 and 3.1, despite their limitations, contribute to enriching our knowledge about what conditions make provable inference scaling laws possible.
>
>
>
> ## Tentative revisions
>
> Motivated by your feedback, we plan to make the following revisions to make the strengths and limitations of Assumptions 2.1 and 3.1 clearer to the readers:
>
> - In Line 58, we will replace "two natural (and arguably minimal) assumptions" with simply "two assumptions".
>
> - In Lines 64 - 65 (the second paragraph of "main contributions" in introduction), we will add one sentence to state our assumption for the league algorithm more explicitly: "there exist correct solutions whose average win rates against a distribution of opponents are higher than that of any incorrect solution".
>
> - In Line 71, we will slightly rephrase the last sentence of introduction, to faithfully acknowledge the gap between theory and practice. A tentative version is as follows: "Although the technical assumptions in our theories can seem strong from a more practical perspective, our empirical results confirm that the proposed algorithms, developed based on the theoretical insights, indeed perform well and demonstrate outstanding scaling properties across diverse LLMs and datasets."
>
> - In Lines 112 - 114 (following the statement of Assumption 2.1), we will remark that $p_{comp} > 0.5$ is assumed for any pair of solutions, which is a strong assumption and thus will motivate our development of the league algorithm (and its theory) in the next section. Such a remark indeed resembles the beginning of Section 3 in the current manuscript.
>
> - In Lines 177 - 181 (following the statement of Assumption 3.1), we will remark that Assumption 3.1 precludes the existence of an adversarial, incorrect candidate that achieves a higher average win rate than any correct solution does.
>
> We'd love to hear your feedback on whether any further improvement can be made.
>
> ---
>
>
> # Other concerns
>
>
> **In experiments, there are problems that do not satisfy the assumptions, and the algorithms do not achieve provable inference scaling laws on such problems or datasets.**
>
> We believe that this is reasonable and well expected, since there are always *counterexamples* to an *assumption*.
> This is also true for other methods like majority voting or Best-of-N,
> whose provable inference scaling laws rely on assumptions that fail on certain problems and datasets in reality
> (cf. Lines 42 - 52, Lines 338 - 349, and the references therein).
>
> Indeed, a large part of our empirical study has been dedicated to understanding the empirical success and failure of the proposed algorithms, and in particular, to what extent the proposed technical assumptions hold true in reality.
> This includes Lines 251 - 308 for the knockout algorithm, Lines 316 - 328 for the league algorithm, and the corresponding figures of experiment results.
> For example, if a problem (e.g., "solve the P versus NP problem") is simply beyond the LLM's capabilities, then the LLM will effectively have $p_{gen} = 0$ or $p_{comp} \le 0.5$ on this problem, which violates Assumption 2.1 required by Theorems 2.2 and 2.3.
> In such cases, it is no surprise that the knockout algorithm (or other test-time scaling methods) fails to solve this problem.
>
>
>
> **The concept of "provable inference scaling law" in Definition 1.1 is not only a property of the algorithm but also of the input problem, which feels less useful than a definition that does not involve the data.**
>
> To the best of our understanding,
> involving the data (i.e., input problem) in the formal definition of "provable inference scaling law" is reasonable and perhaps inevitable.
> After all, given the capabilities and limitations of today's LLMs, we do not expect that there exists an LLM-based algorithm that can solve *any* possible input problem by scaling up its test-time compute.
> In other words, the feasibility of a provable inference scaling law must rely on certain assumptions not just about the LLM and algorithm, but also about the input problem.
> We are uncertain about what a definition that doesn't involve the data should look like,
> and would appreciate further hints on this aspect.
>
>
>
> **Comparison between the proposed algorithms and the baseline of majority voting, "the authors only count the compute to generate the candidates but neglect the compute for aggregation".**
>
> We would like to clarify that **the costs of both generation and aggregation stages have been taken into account in our comparison** between the knockout algorithm and majority voting, although we choose to use $N$ as the X-axis in Figure 2 for clean visualization.
> In particular, we highlight in **Line 244** that the knockout algorithm requires a total of $5 \times N$ or $3 \times N$ (depending on the experiment settings) LLM calls for solving a problem, while majority voting requires only $N$ LLM calls;
> to improve the clarity of our presentation, we plan to add this message to the caption of Figure 2 in our revision.
> Even when using the total number of LLM calls (rather than $N$) as the cost metric,
> the knockout algorithm still demonstrates advantages over majority voting in terms of convergence speed and ceiling of the accuracy curves,
> as explained in **Lines 246 - 250**.
>
>
>
>
>
> ---
>
> Thank you again for your review, which helps us a lot in improving the current work!
> We hope that our responses have provided satisfactory answers to your questions,
> but do let us know if you have any further concern that we can address.

---

> > ### Comment · Reviewer_PgPW · 2025-08-02
> >
> > I thank the authors for their detailed rebuttal.
> >
> > The authors' clarification have addressed my concerns regarding the fairness of the comparison between the proposed algorithms and the majority voting baseline. The proposed revisions on the assumptions also look good to me.
> >
> > I recognize the challenge of formulating weaker assumptions or providing a better definition. I agree with the authors that the proposed assumptions represent an improvement over those required by BoN, although this improvement appears relatively minor.

---

> > > ### Author Response · Authors · 2025-08-04
> > >
> > > Thank you for your reply!
> > >
> > > We're glad to know that your concerns have been resolved,
> > > and appreciate that you acknowledge the challenges of proposing a better formal definition for "provable inference scaling law", or weaker assumptions that make it possible to achieve such a property, than those proposed in our work.
> > >
> > > We trust that your finalized review of this work will be a comprehensive and fair evaluation that takes into account our overall contributions to the area of LLM test-time scaling, including:
> > > 1. the identified technical assumptions, which seem minimal in some senses (e.g., $p_{gen} > 0$ and $p_{comp} > 0.5$ at the tightest possible threshold) but also strong in others (e.g., precluding the existence of an adversarial candidate);
> > > 2. the theoretical insights and practical algorithms (with provable guarantees) built upon such assumptions; and
> > > 3. the experiments that extensively demonstrates the empirical advantages of our proposed algorithms compared to baselines, as well as investigates their limitations in practice.
> > >
> > > As always, we're open to further discussions if you need more information before making your final decision :)

---

> > > > ### Comment · Reviewer_PgPW · 2025-08-04
> > > >
> > > > Although the authors have not fully addressed my concerns regarding the assumptions, which I recognize might have been challenging given the limited rebuttal period, the proposed revisions and clarifications related to these assumptions enhance the rigor and professionalism of the paper. Therefore, I have raised my score to reflect these improvements. However, due to my remaining concerns about the assumptions, I am unable to further increase my score.

---

> > > > > ### Author Response · Authors · 2025-08-05
> > > > >
> > > > > Thank you for updating your evaluation!
> > > > >
> > > > > We acknowledge that while our proposed technical assumptions already represent an improvement over those required by BoN,
> > > > > they are not the “ultimate” solution.
> > > > > Our primary focus in this work is to rigorously reveal some essential principles about LLM test-time scaling
> > > > > (e.g., how to amplify the overall success probability up to 100% based on pairwise comparisons that are merely better than random guess),
> > > > > which will be helpful for the community to discover the “ultimate” solution together.
> > > > > It is indeed intriguing future work to identify even weaker assumptions that make it possible to achieve provable inference scaling laws,
> > > > > and we hope that our work makes a good contribution towards this direction.
> > > > >
> > > > > Thank you again for your feedback and discussion!

---

### Decision · Program_Chairs · 2025-09-17

**Decision:**

Accept (poster)

**Comment:**

Test time compute is increasingly being used to improve the accuracy of LLMs. For example, a very simple method is to over-sample several possible answers, and then output the majority vote. This paper proposes two cleverer methods inspired by tournament scoring. Both methods involve having different answers "compete" with each other, with the winning answer moving onto the next round. In addition to providing experimental results, the paper analyzes the success probability of each method as the number of answers considered is increased.

Reviewers liked that the paper provides formal guarantees of the method's success rate, though they do note that the assumptions needed for these guarantees to be true are quite strong, and this limitation is not sufficiently addresses in the paper. The reviewers note that the empirical results are convincing, although there is some concern that the authors didn't choose enough baselines to compare against.

Overall, I think the proposed methods are interesting and the paper is enjoyable to read. I recommend its acceptance.

However, in their camera ready version, the authors should make sure to add a longer discussion of Assumptions 2.1 and 3.1 and their limitations.